# Hypercaloric diet models do not develop heart failure, but the excess sucrose promotes contractility dysfunction

Amanda Martins Matias[1], Priscila Murucci Coelho[1], Vinícius Bermond Marques[2], Leonardo dos Santos [2], Aricia Leone Evangelista Monteiro de Assis[3], Breno Valentim Nogueira[3], Ana Paula Lima-Leopoldo[1,4], André Soares Leopoldo [1,4] *

1 Postgraduate Program in Nutrition and Health, Center of Health Sciences, Federal University of Espírito Santo, Vitória, Espírito Santo, Brazil, 2 Center of Health Sciences, Department of Physiological Sciences, Federal University of Espírito Santo, Vitória, Espírito Santo, Brazil, 3 Center of Health Sciences, Department of Morphology, Federal University of Espírito Santo, Vitória, Espírito Santo, Brazil, 4 Department of Sports, Center of Physical Education and Sports, Federal University of Espírito Santo, Vitória, Espírito Santo, Brazil

* andre.leopoldo@ufes.br

**Data Availability Statement:** All relevant data are within the manuscript.

## Abstract

Several diseases are associated with excess of adipose tissue, and obesity is considered an independent risk factor for the development of cardiac remodeling and heart failure. Dietary aspects have been studied to elucidate the mechanisms involved in these processes. Thus, the purpose was the development and characterization of an obesity experimental model from hypercaloric diets, which resulted in cardiac remodeling and predisposition to heart failure. Thirty- day-old male Wistar rats (n = 52) were randomized into four groups: control (C), high sucrose (HS), high-fat (HF) and high-fat and sucrose (HFHS) for 20 weeks. General characteristics, comorbidities, weights of the heart, left (LV) and right ventricles, atrium, and relationships with the tibia length were evaluated. The LV myocyte cross sectional area and fraction of interstitial collagen were assayed. Cardiac function was determined by hemodynamic analysis and the contractility by cardiomyocyte contractile function. Heart failure was analyzed by pulmonary congestion, right ventricular hypertrophy, and hemodynamic parameters. HF and HFHS models led to obesity by increase in adiposity index (C = 8.3 ± 0.2% *vs*. HF = 10.9 ± 0.5%, HFHS = 10.2 ± 0.3%). There was no change in the morphological parameters and heart failure signals. HF and HFHS caused a reduction in times to 50% relaxation without cardiomyocyte contractile damage. The HS model presented cardiomyocyte contractile dysfunction visualized by lower shortening (C: 8.34 ± 0.32% vs. HS: 6.91 ± 0.28), as well as the $Ca^{2+}$ transient amplitude was also increased when compared to HFHS. In conclusion, the experimental diets based on high amounts of sugar, lard or a combination of both did not promote cardiac remodeling with predisposition to heart failure under conditions of obesity or excess sucrose. Nevertheless, excess sucrose causes cardiomyocyte contractility dysfunction associated with alterations in the myocyte sensitivity to intracellular $Ca^{2+}$.

**Funding:** This study was supported by the Brazilian National Council for Scientific and Technological Development – CNPq (grant number: 402090/2016-0) to ASL and FAPES (grant number: 590/2019). The funders had no role in study design, data collection and analysis, decision to publish, or preparation of the manuscript.

**Competing interests:** The authors have declared that no competing interests exist.

## Introduction

Obesity is a disease associated with several disorders and the intensity and duration of this condition are associated with cardiac remodeling with presence or absence of heart failure [1–4]. The excess of adipose tissue requires increased metabolic demands, which leads to chronic volume overload and, consequently, cardiac remodeling. In addition, changes in cardiovascular hemodynamics, can modify the cardiac morphology and function. Thus, the association of obesity with comorbidities, some neurohormonal and metabolic alterations may result in heart failure [5, 6]. In this context, dietary aspects have been studied to explain the possible mechanisms related to the emergence of this condition, which can modulate cardiovascular risk factors.

High-fat intake is not only related to lipid metabolism, but the type of ingested fat can also influence insulin resistance and promote alterations in blood pressure [7]. Diets with high intakes of simple carbohydrates may result in increased heart exposure to insulin, which activates cardiac protein synthesis and may promote left ventricular hypertrophy [8, 9].

Studies have shown that obesity induced by hypercaloric diets promotes cardiac remodeling with changes in morphology and/or cardiac function [10, 11]. Evidence from experimental studies indicates that high-fat feeding promotes cardiac contractile function damage [12, 13]. In our laboratory, myocardial dysfunction was demonstrated in basal conditions and physiological cardiac remodeling in obese rats [14–16]. In contrast, results on the effects of diets with high simple carbohydrate content (*sucrose and/or fructose*) are inconsistent. Sharma et al. [9] observed that hypertensive animals fed by fructose diet for eight weeks resulted in an increase in LV wall thickness and mortality, while Salie et al. [17] demonstrated cardioprotective effect after ischemia reperfusion in *Wistar* rats, using diet with sucrose supplementation for 16 weeks. Moreover, high-fat diets associated with simple carbohydrates have demonstrated cardiac damage, ventricular hypertrophy, interstitial fibrosis, increased rigidity and impairment in cardiac relaxation [17, 18].

Despite much research and evidence of cardiovascular damage caused by high-fat and high-sucrose diets, it is still unclear whether these diets, isolated or combined, promote cardiac remodeling with predisposition to heart failure. Thus, this study aimed to investigate the cardiac remodeling process in an experimental model induced by different types of hypercaloric diets (high-fat (lard), high-sugar and the combination of both) and their effects in the cardiac function. The hypothesis was that hypercaloric diets would promote cardiac remodeling, cardiovascular damage and predispose to heart failure, being most evident in the HFHS model.

## Material and methods

### Animal care and experimental design

Thirty-day-old male *Wistar* rats ($\cong$110 g) obtained from the Animal Quarters of the Federal University of Espírito Santo (Brazil) were individually caged and subjected to different dietary regimens. All animal experiments were approved by the Ethics Review Committee of Federal University of Espírito Santo (CEUA-UFES 08/2016) and conducted in accordance with current Brazilian laws.

Rats were randomly assigned in control diet (C; n = 12), high-sugar diet (HS; n = 14), high-fat diet (HF; n = 13) and high-fat and high-sugar diet (HFHS; n = 13). All animals had free access to water and chow (40 g/day). To analyze whether dietary-induced obesity was associated with alterations in nutritional behavior, food consumption (FC) was measured daily. Calorie intake (CI) was calculated weekly by the average weekly FC × dietary energetic density. Feed efficiency (FE), the ability to transform consumed calories into body weight, was

determined by following the formula: mean body weight gain (g)/total calorie intake (kcal). The HS group had water supplemented with sugar (300 g/l) in alternate weeks. For the calculation of the caloric intake of the HS group, the caloric energy from the water supplemented with sugar was also quantified (1.2 kcal/mL consumed).

The experimental diets provided sufficient amounts of protein, vitamins and minerals according to the Nutrient Requirements for Laboratory Animals. The diets used in the current study were formulated by Nutriave Alimentos® (Vitória, Espírito Santo, Brazil) [19]. The feed ingredients were blended, homogenized and extruded (Extru-Tech Extruder, Model E-750, Sabetha, KS, USA) in the form of pellets. Then, the pellets were dried on a horizontal conveyor dryer (20 minutes, temperature: ±70˚C). The composition (g/kg) and nutrients for each experimental diet (%) are described in Table 1. The duration of the experimental protocol was 20 consecutive weeks.

## Characterization of obesity

The adiposity index, used to assess obesity, was calculated by adiposity index [body fat (BF)/ final body wt] ×100. BF was measured from the sum of the individual fat pad weights as follows: BF = epididymal fat + retroperitoneal fat + visceral fat.

## Metabolic and hormonal measurements

After 20 weeks, the animals were subjected to 12–15 h of fasting, and blood samples were collected in dry tubes. The serum was separated by centrifugation at 10,000 rpm for 10 min.

**Table 1. Composition and nutritional values of diets.**

| Components (g/kg) | Diets | | | |
|---|---|---|---|---|
| | C | HS | HF | HFHS |
| Corn | 200 | 200 | 180 | 80 |
| Rice | 200 | 200 | 200 | 200 |
| Bone meal | 120 | 120 | 120 | 120 |
| Sugar | - | 100 | - | 100 |
| Soy oil | 75 | 75 | - | - |
| Lard | - | - | 200 | 200 |
| Gluten | 200 | 200 | 200 | 200 |
| Salt | 3.5 | 3.5 | 3.5 | 3.5 |
| Mineral Mix** | 35 | 35 | 30 | 30 |
| Vitamin Mix** | 16.5 | 16.5 | 16.5 | 16.5 |
| Inert Material*** | 150 | 50 | 50 | 50 |
| Total (g) | 1000 | 1000 | 1000 | 1000 |
| Nutrient Composition (%) | | | | |
| Protein | 24.8 | 21.8 | 17.8 | 19.2 |
| Carbohydrate | 49.6 | 52.3 | 44.6 | 43.4 |
| Lipids | 25.6 | 25.9 | 37.6 | 37.4 |
| Energy Density (Kcal/g) | 3.55 | 3.65 | 4.59 | 4.49 |

Diets. C: normal rodent chow; HS: High-sugar; HF: high-fat; HFHS: high fat and high sugar. In order to calculate the caloric intake of HS, the caloric value of the sugar diet (3.65 kcal/g) plus the caloric value of water intake with sugar (1.2 kcal/ml) was computed ** Vitamin and Mineral Mix: vit. A, vit. C., vit. D3, vit. E, vit. K3, vit. Complex B, pantothenic acid, folic acid, biotin, choline; selenium, iron, copper, manganese, iodine, zinc, cobalt, calcium, and phosphorus.

*** Bentonite: inert material, with no nutritional value and calories.

(Heraeus Megafuge 16R Centrifuge, Thermo Scientific, Massachusetts, USA) and stored at −80 ˚C for subsequent analysis (Coldlab Ultra Freezer CL374-86V, Piracicaba, São Paulo, Brazil). Serum glucose concentration was measured using specific kit (Bioclin Bioquímica®, Belo Horizonte, Minas Gerais, Brazil and Synermed do Brasil Ltda., São Paulo, Brazil) and analyzed by automated biochemical equipment BS-200 (Mindray do Brasil-Comércio and Distribuição de Equipamentos Médicos Ltda., São Paulo, Brazil). Insulin was determined using an enzyme-linked immunosorbent assay (ELISA) using specific kit (Linco Research Inc., St. Louis, MO, USA). The reading was carried out using a microplate reader (Asys Expert Plus Microplate Reader, Cambourne, Cambridge, UK).

## Cardiac remodeling assessment

The cardiac remodeling process was assessed by determination of heart weight (HW), left ventricle, HW and LV/tibia length ratios. At the end of the experimental protocol, the animals received an intraperitoneal injection (IP) of sodium heparin (1000 U/Kg/IP; Heparamax-s, Blau Pharmaceutic S.A., São Paulo, Brazil) for cardiomyocyte analysis. After 30 minutes, rats were anesthetized with ketamine (90 mg/kg, IP) plus xylazine (10 mg/kg, IP) and euthanized by decapitation. Subsequently, their chests were opened by mid-thoracotomy, and the heart, ventricles, fat pads, and tibia were separated, dissected, weighed or measured.

**Hemodynamic measurements.**   After 20 weeks of experimental protocol, the rats (n = 6 per group) were anesthetized intraperitoneally (IP) with urethane (Carbamic acid ethyl ester; 1.2 g/kg/IP injection; Sigma-Aldrich, USA) and submitted to catheterization surgery. LV hemodynamic data were obtained from right carotid artery using a micromanometer (Mikro-Tip$^{TM}$, SPR 320, USA). Systolic and diastolic blood pressure (SBP and DBP), heart rate (HR); LV systolic and end-diastolic pressure (LVSP and LVEDP); maximum positive (+dP/dt$_{max}$) and negatives (-dP/dt$_{max}$) derivatives of LV pressure, LV relaxation time constant (TAU) were acquired and analyzed using a computer (Biopac System, USA).

**Histological analysis.**   The LV transverse sectional area of animals from each group was fixed in phosphate-buffered 4% paraformaldehyde (7.4 pH) and embedded in paraffin. Sections of 6 μm were obtained and stained with hematoxylin-eosin (HE) and picrosirius red stain to determine the myocyte cross-sectional area and to collagen volume fraction, respectively.

**Cardiomyocyte contractile function.**   Under anesthesia as described above, the hearts from rats were quickly removed by median thoracotomy, then enzymatically isolated [20]. The isolated cells were placed in an experimental chamber with a glass coverslip base mounted on the stage of an inverted microscope (IonOptix, Milton, USA) coupled with an edge detection system with a 40× objective lens (Nikon Eclipse–TS100, USA). Cells were immersed in Tyrode's solution containing 1.8 mM CaCl$_2$ and field stimulated at 1 Hz (20V, 5ms duration square pulses). Fractional shortening and the times to 50% contraction and relaxation were measured.

**Intracellular Ca$^{2+}$ measurements.**   Myocytes were loaded with 1.0 μM Fura2-acetoxymethyl (AM) ester (Molecular Probes, USA). Ca$^{2+}$ transient amplitude was reported as F/F0, where F is the maximal fluorescence intensity average measured at the peak of [Ca$^{2+}$]i transients, and F0 is the baseline fluorescence intensity measured at the diastolic phase of [Ca$^{2+}$]i transients. We also analyzed the period of time until the Ca$^{2+}$ transient peak and 50% Ca$^{2+}$ decay were reached.

## Heart failure

Heart failure was considered when three criteria were met: *Pulmonary congestion*: a) percentage of lung humidity above 80%; b) lung/body weight (BW) ratio greater than lung/BW

approximately 2-fold in relation to C [21]; c) right lung/BW above 5 mg/g [22]. *RV hypertrophy*: characterized by the ratio of the weight of RV adjusted by BW above 0.8 mg/g [23]. Elevated *LVEDP*: LV end diastolic pressure (LVEDP) greater than 15 mmHg [24, 25]; and *Systolic dysfunction*: maximum positive value of the first derivative of LV systolic pressure above 7000 mmHg [22]. After removal of the lung tissue, wet weight (WW) and dry weight (DW) were determined before and after drying the samples at 60°C for 48 hours. Water content (%$H_2O$) was estimated using the follow formula: %H2O = [(WW—DW)/WW] × 100.

## Statistical analysis

Data were reported as mean ± standard error of the mean (SEM) and analyzed using one-way analysis of variance (ANOVA) followed by the Tukey *post hoc*, at a 5% significance level.

## Results

After 20 weeks, HF rats demonstrated a final body weight 22% and 20.6% greater than C and HS, respectively (Table 2). The HF and HFHS diets promoted a substantial elevation of visceral and retroperitoneal fat pads compared to C and HS diets. Specifically, HF and HFHS rats showed an elevation of 62% and 43% in body fat, respectively, when compared to C. In addition, the adiposity index was significantly greater in these groups (31.3% and 22.9%) than in C. In relation to HS, there was an increase in body fat of 68% and 48.7% in HF and in HFHS, respectively.

The nutritional profile of rats is summarized in Table 2. The C rats had an approximately 17.9%, 29.6% and 37.6% greater daily food consumption (g) than the HF, HFHS and HS groups, respectively, but the daily caloric intake was higher in the HS group in relation to the C and HFHS groups (HS: 92.1 ± 2.1 *vs.* C: 79.2 ± 2.6 and HFHS: 77.3 ± 1.8 kcal/day, p < 0.05).

**Table 2. General characteristics.**

| Variables | Experimental Groups | | | |
|---|---|---|---|---|
| | C | HS | HF | HFHS |
| IBW (g) | 107 ± 3 | 110 ± 3 | 111 ± 3 | 110 ± 4 |
| FBW (g) | 533 ± 17 | 538 ± 13[#] | 649 ± 34[*] | 616 ± 22 |
| BW gain (g) | 426 ± 17 | 428 ± 12[#] | 538 ± 32[*] | 506 ± 20 |
| Epididymal fat pad (g) | 11.2 ± 0.6 | 10.5 ± 0.7 | 13.1 ± 0.9 | 13.5 ± 1.0 |
| Visceral fat pad (g) | 11.4 ± 0.6 | 10.6 ± 0.6[#&] | 18.5 ± 1.4[*] | 15.9 ± 1.4[*] |
| Retroperitoneal fat pad (g) | 21.9 ± 1.0 | 21.8 ± 2.0[#&] | 40.4 ± 4.5[*] | 34.4 ± 2.2[*] |
| Body fat (g) | 44.5 ± 1.6 | 42.9 ± 0.6[#&] | 72.0 ± 6.3[*] | 63.8 ± 4.3[*] |
| Adiposity index (%) | 8.3 ± 0.2 | 7.9 ± 0.3[#&] | 10.9 ± 0.5[*] | 10.2 ± 0.3[*] |
| Food consumption (g/day) | 22.3 ± 0.7 | 16.2 ± 0.5[*#] | 18.9 ± 0.7[*] | 17.2 ± 0.4[*] |
| Caloric intake (kcal/day) | 79.2 ± 2.6 | 92.1 ± 2.1[*&] | 86.9 ± 3.5 | 77.3 ± 1.8[α] |
| Feed efficiency (%) | 3.84 ± 0.08 | 3.32 ± 0.05[*] | 4.39 ± 0.09[*#] | 4.64 ± 0.09[*&] |
| Glucose (mg/dL) | 108 ± 2 | 112 ± 3 | 115 ± 4 | 115 ± 3 |
| Insulin (ng/mL) | 1.86 ± 0.13 | 1.77 ± 0.15 | 2.19 ± 0.16 | 2.39 ± 0.23 |

Data are presented as the mean ± SEM. Control diet—(C; n = 12); high-sugar diet—(HS; n = 14); high-fat diet—(HF; n = 13), and high-fat and high-sugar diet (HFHS; n = 13). IBW: initial body weight; FBW: final body weight; BW: body weight. One-way ANOVA for independent samples followed by Tukey *post hoc* test. p < 0.05 *vs.* [*] C

[#] HF *vs.* HS

[&] HFHS *vs.* HS

[α] HF *vs.* HFHS.

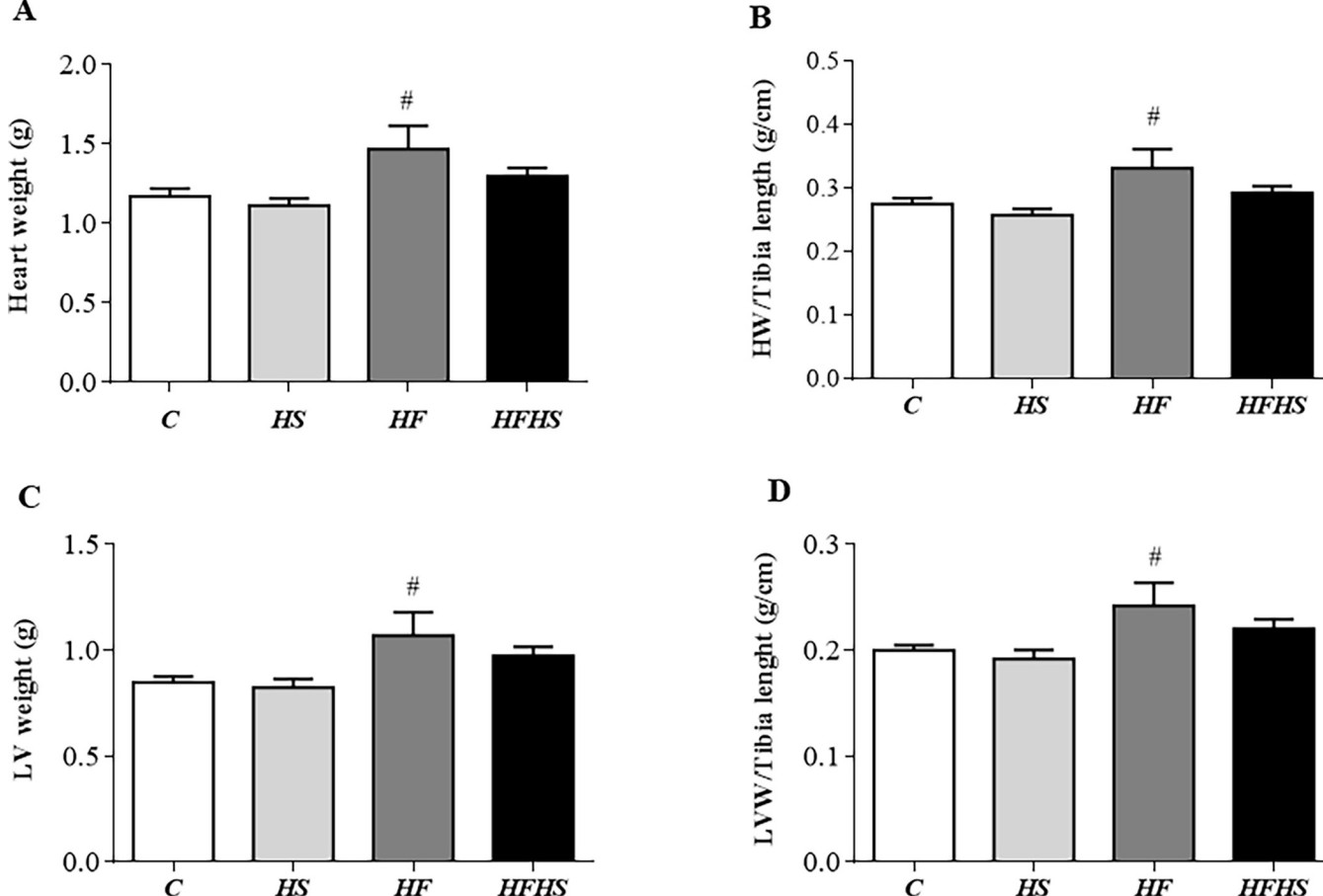

**Fig 1. Effect of different diet composition on cardiac remodeling.** Data are shown as mean ± SEM. Control diet—(C; n = 5); high-sugar diet—(HS; n = 8); high-fat diet—(HF; n = 5), and high-fat and high-sugar diet (HFHS; n = 6). HW: heart weight; LW: left weight. p < 0.05 vs. # HF vs. HS. One-way analysis of variance (ANOVA) followed by the Tukey *post hoc* test.

In addition, there was a difference in FC between rats fed the HF diet compared with the HS group since rats fed the HS diet consumed significantly less food (HG: 16.2 ± 0.5 *vs.* HF:18.9 ± 0.7; p < 0.05). Furthermore, HF presented a 12.4% increase in caloric intake over HFHS (p < 0.05). There was no difference in the caloric intake of C compared to HF and HFHS (p > 0.05). While the feed efficiency (%) was higher in the HF (14.3%) and HFHS groups (20.8%) than in C (Table 2), there was a lower feed efficiency in HS rats than C (HS: 3.32 ± 0.05 *vs.* C: 3.84 ± 0.08; p < 0.05).

In relation to cardiac remodeling, there were no differences in the parameters analyzed in groups HS, HF and HFHS when compared to C (Fig 1). However, the absolute heart and LV weights were significantly elevated in HF rats in relation to HS group (Fig 1A and 1B), representing an increase of 32% and 29%, respectively. In addition, these rats also showed elevation of HW and LVW-to-tibia length ratios when compared to HS rats (Fig 1C and 1D). Additionally, the histological analysis from LV samples revealed that CSA and interstitial collagen fraction (%) were similar among the groups (Fig 2A and 2B). There was no statistical difference between the experimental groups for the lung/FBW, right lung/FBW and RV/FBW ratios, as well as for the lung water content (%) (Fig 3A–3D).

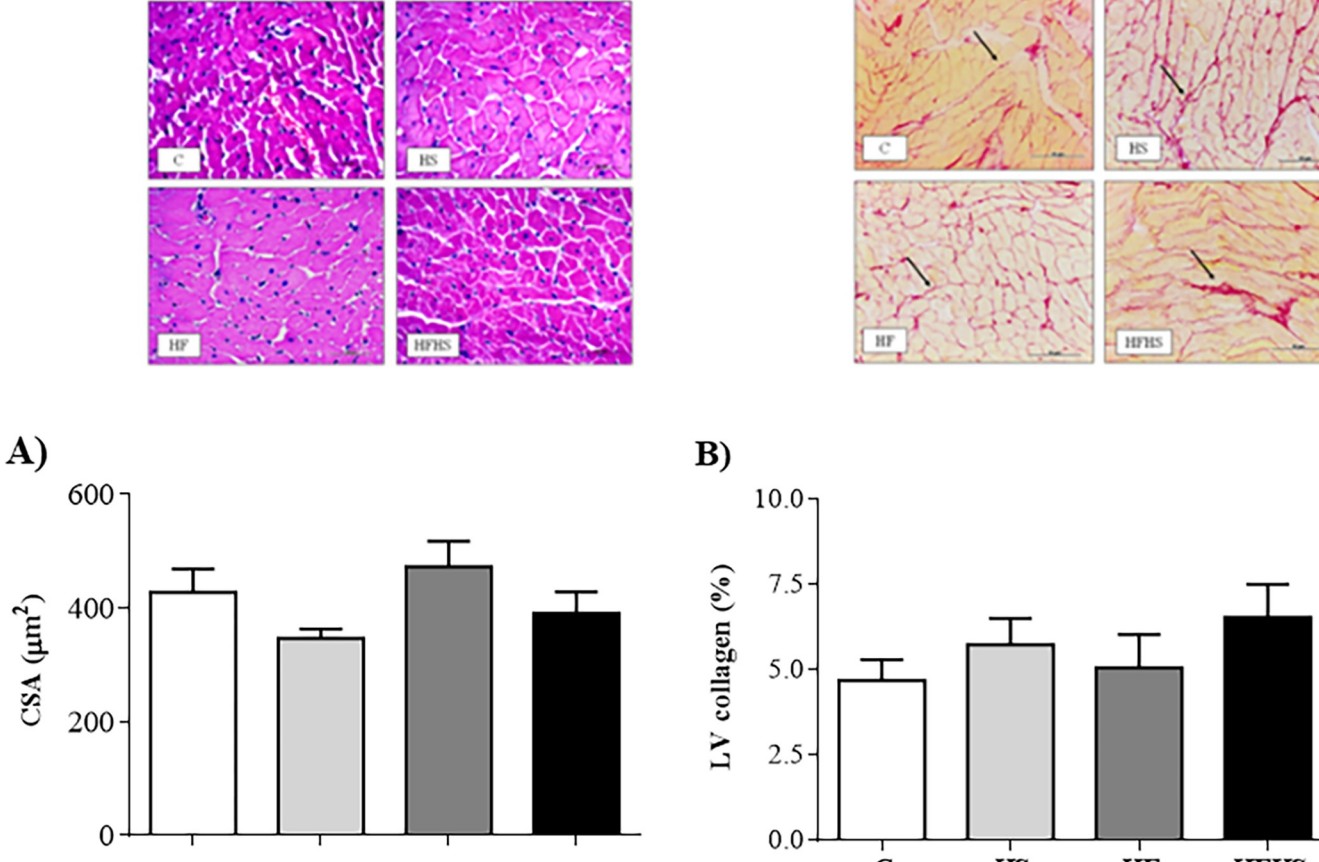

**Fig 2. Histological study in myocardium.** Control diet—(C; n = 5); high-sugar diet—(HS; n = 8); high-fat diet—(HF; n = 5), and high-fat and high-sugar diet (HFHS; n = 6) after 20 weeks. (A) cross sectional area (CSA). (B): interstitial collagen of myocardium; representative picrosirius red-stained left ventricle (LV) section. Arrows: interstitial collagen. Data are shown as mean ± SEM. One-way analysis of variance (ANOVA) followed by the Tukey *post hoc* test. p < 0.05 vs. * C; & HFHS vs. HS; §HFHS vs. HF.

The different types of hypercaloric diets did not promote hemodynamic alterations (Table 3). In addition, the fractional shortening was significantly reduced by 20% and 15% in the HS group in relation to C and HFHS, respectively (Fig 4B). Although all groups showed lower values of time to 50% contraction in relation to C, there only was statistical difference between HF and C (Fig 4C). The time to 50% relaxation were reduced in HS when compared to HF and HFHS, respectively (Fig 4D), resulting in impaired of cardiomyocyte relaxation. Moreover, HF and HFHS groups had reduced times to 50% relaxation in relation to C (Fig 4D). $Ca^{2+}$ transient amplitude was increased in HS compared to HFHS (Fig 4E). In contrast, no differences were observed in times to peak and 50% $Ca^{2+}$ decay among other groups (Fig 4F and 4G).

## Discussion

Obesity is associated with structural and functional changes in the heart [1]. However, the HS did not cause morphological cardiac remodeling. Chess and Stanley [26] reported that high sugar intake leads to hyperglycemia in blood circulation and cardiotoxic effects. Increased glycemia leads to elevated serum insulin levels, which in cardiac tissue induce elevated protein synthesis of cardiomyocytes and, consequently, left ventricular hypertrophy. Sharma et al. [9]

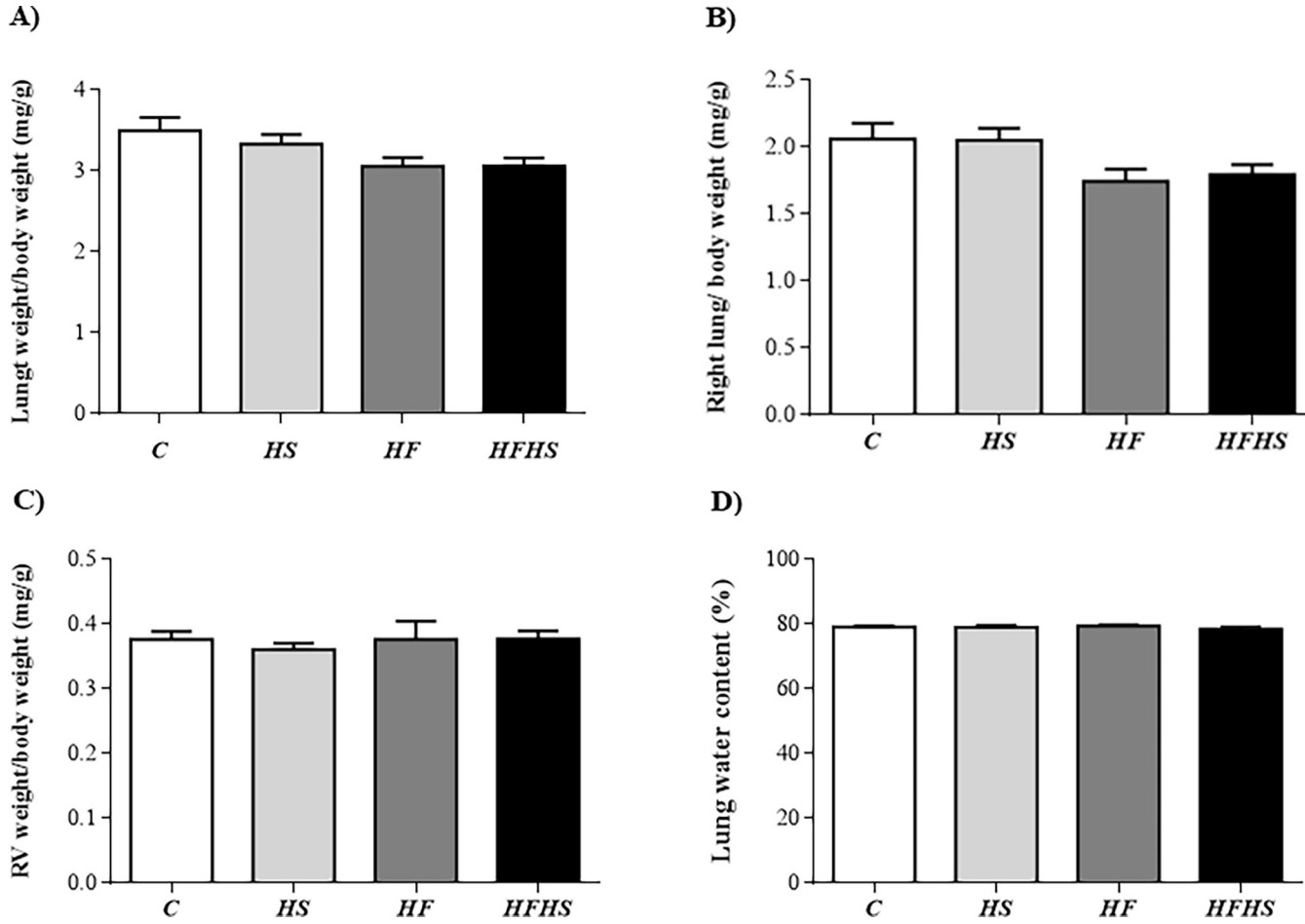

**Fig 3. Effect of different diet composition on parameters of heart failure.** Data are shown as mean ± SEM. Control diet—(C; n = 12); high-sugar diet—(HS; n = 14); high-fat diet—(HF; n = 13), and high-fat and high-sugar diet (HFHS; n = 13). RV: right ventricle. (A) Lung weight/body weight ratio. (B) Right lung weight/body weight ratio. (C) RV weight/body weight. (D) Lung water content (C = 5; HS = 8; HF = 5, and HFHS = 6). One-way analysis of variance (ANOVA) followed by the Tukey *post hoc* test.

proposed that the ingestion of carbohydrates, in particular sugar, associated or not with blood pressure overload, can cause LV hypertrophy, via insulin receptor stimulation and activation of Akt/mTOR pathways. These findings suggest that, in the HS model, there was no need for elevated circulating insulin in response to increased glycemic levels, since the animals remained euglycemic.

The HF model did not cause cardiac remodeling, although the animals showed higher heart weight and heart/tibia length ratio compared to C, however, this difference was not significant (p = 0.08). In addition, although the myocyte CSA of this group had a higher value in relation to C, this increase was also not significant (p = 0.07). However, the HF diet promoted cardiac remodeling in relation to HS group (Fig 1A and 1B). In relation to the HFHS, no changes in heart weight, CSA and myocardial collagen fraction were observed, which shows that this treatment did not lead to cardiac remodeling. These findings are divergent from Poudyal et al. [27], who visualized cardiac remodeling with LV hypertrophy and increased collagen fraction in *Wistar* rats after 32 weeks of experimental protocol. The authors reported that these changes are related, among other factors, to the aging process. Another mechanism is related to leptin

**Table 3. Left ventricular hemodynamics measurements.**

| Variables | Experimental Groups | | | |
|---|---|---|---|---|
| | C | HS | HF | HFHS |
| SBP (mmHg) | 97.8 ± 3.5 | 98.9 ± 4.5 | 92.7 ± 3.8 | 98.8 ± 3.5 |
| DBP (mmHg) | 58.5 ± 4.5 | 61.1 ± 5.2 | 56.5 ± 2.7 | 59.6 ± 5.1 |
| HR (bpm) | 288 ± 10 | 314 ± 17 | 321 ± 11 | 328 ± 14 |
| LVSP (mmHg) | 99.5 ± 2.9 | 95.8 ± 4.5 | 93.4 ± 3.9 | 100 ± 3 |
| LVDP (mmHg) | 4.02 ± 1.6 | 2.17 ± 0.81 | 2.19 ± 0.85 | 2.8 ± 0.79 |
| +dP/dt$_{máx}$ (mmHg/s) | 6568 ± 471 | 7301 ± 1045 | 6157 ± 720 | 8022 ± 652 |
| -dP/dt$_{máx}$ (mmHg/s) | -6769 ± 389 | -6485 ± 725 | -6375 ± 661 | -7749 ± 416 |
| Tau (s) | 0.014 ± 0.003 | 0.011 ± 0.001 | 0.010 ± 0.002 | 0.011 ± 0.001 |

Data are presented as the mean ± SEM (n = 6 animal per group). Control diet—(C); high-sugar diet—(HS); high-fat diet—(HF), and high-fat and high-sugar diet (HFHS). SBP: systolic blood pressure; DBP: diastolic blood pressure; HR: heart rate; LV: left ventricle; LVSP: LV systolic pressure; LVEDP: LV end-diastolic pressure; +dP/dt$_{máx}$: maximum positive derivative of LV pressure; -dP/dt$_{máx}$: maximum negative derivative of LV pressure; Tau: LV relaxation time constant. One-way ANOVA for independent samples followed by Tukey *post hoc* test.

that mediates the process of cardiac hypertrophy in parts by the mitogen-activated protein kinase (MaPK p38), which regulates various cellular processes.

Specifically, the HS did not promote cardiac remodeling, however, it was able to prevent the elevation on heart and LV weights in the HFHS, interfering in the effect of HF diet. According to literature, high sucrose intake [68% total energy sucrose diet, 69% total mass sucrose diet, or 20% sucrose solution] induced hyperinsulinemia without alterations in plasma glucose level in rats and mice [28–30]. Interestingly, our HS rats have not demonstrated alterations in glucose and insulin levels (Table 2). Several mechanisms have been postulated, including the prohypertrophic effects of insulin, insulin growth factor-1, and insulin resistance [31]. Initially the circulating insulin levels are increased, directly stimulating cardiomyocyte growth [32] and indirectly via binding to the insulin growth factor-1 receptor [33]. Chess et al. [26], analyzing the effects of dietary extremes (high carbohydrate and fat intake) on the remodeling process and heart failure, reported that high sugar intake causes hyperglycemia in the bloodstream and cardiotoxic effects. Increased glycemia leads to elevated serum insulin levels, which in cardiac tissue, induces increased protein synthesis of cardiomyocytes and, consequently, left ventricular hypertrophy. Sharma et al. [9] propose that carbohydrate intake, in particular sugar, associated or not with pressure overload may cause LV hypertrophy, via insulin receptor stimulation and activation of Akt/mTOR, proteins involved in protein signaling pathways. Thus, our results from HS diet suggest that the sugar intake was not able to affect the process of cardiac remodeling, probably due to the absence of hyperinsulinemia and elevation of glucose levels, as well as it was not promote cardiotoxic effect, indicating a cardioprotective effect alone or when associated with the HF diet.

Experimental models that mimic the eating habits of the population have been widely used to elucidate the mechanisms of obesity and cardiovascular disorders [34]. In this sense, there was no significant difference was observed *in vivo* study, indicating that the dietary interventions used did not cause cardiac functional adaptation. In addition, our results demonstrate that obesity models (HH and HFHS) preserved the cardiomyocyte contractile function with a punctual improvement in myocardial relaxation. These results differ from other studies that have shown impairment in contraction and relaxation in experimental obesity models [12, 15, 16, 35]. These authors observed a reduction in the L-type Ca$^{2+}$ channels expression, lower phosphorylation of RyR$_2$, and a decrease in SERCA2a and phospholamban phosphorylation

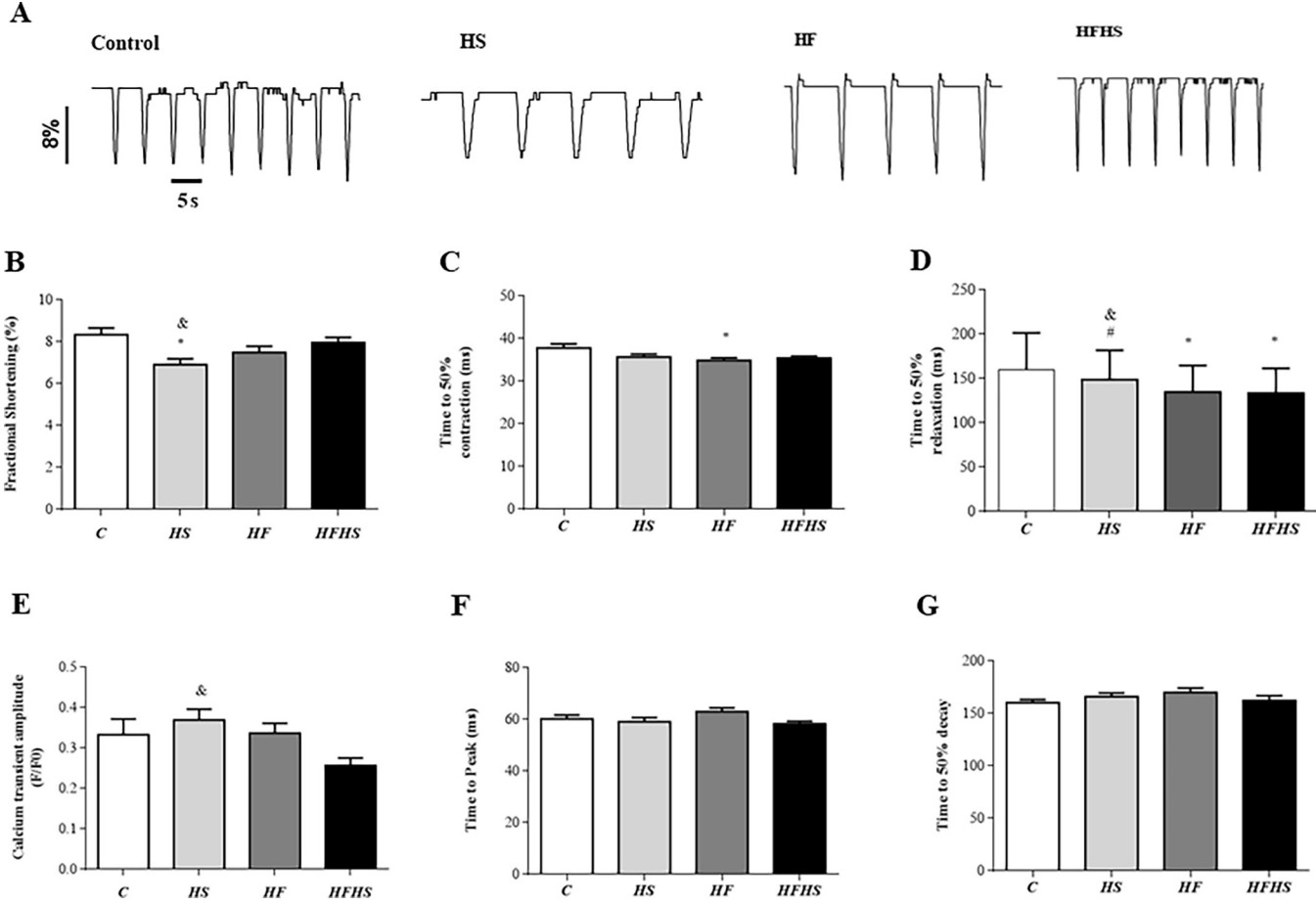

**Fig 4. Effect of different diet composition on contractile function and calcium transients of left ventricular cardiomyocytes.** Control diet (C; n = 5; cells = 71), high-sugar diet (HS; n = 6, cells = 115), high-fat diet (HF; n = 6, cells = 106) and high-fat and high-sugar diet (HFHS; n = 5, cells = 81). Data are shown as mean ± SEM. (A) Representative contraction traces obtained from the cardiomyocytes of rats: (B) Cell shortening expressed as % of resting cell length. (C) Time to 50% of contraction. (D) Time to 50% of relaxation. (E) Amplitude of transients. (F) Time to peak. (G) Time to from peak transient to half resting value p < 0.05 vs. * C; # HS vs. HF; & HFHS vs. HS. One-way analysis of variance (ANOVA) followed by the Tukey *post hoc* test.

(PLB), respectively, due to the increase of intracellular $Ca^{2+}$ concentration and removal of cytosolic $Ca^{2+}$. However, Fauconnier et al. [36] observed that, in the presence of palmitate, cellular shortening increased by 40%, as well as a 25% increase in the ratio between cellular shortening and the $Ca^{2+}$ transient amplitude, demonstrating that this substrate was able to improve the contractile responses. Thus, to maintain contractile function, the heart requires a continuous and abundant supply of energy, which comes mainly from fatty acids and glucose.

Free fatty acid oxidation is the major source of energy for the myocardium and up to 80% of high-energy phosphates are produced, while the glucose metabolism provides the remaining quantity of energy. Glucose as glycogen are stored to be used during increased metabolic demands such as obesity and diabetes, since the glucose utilization is 20–30% more metabolic efficient than free fatty acid oxidation in producing high-energy phosphates. In this sense, an energetic dysregulation play an important role in the pathophysiology of the failing heart. According Doesn't et al. [37] a possible cause for these metabolic derangements in HF could be related to myocardial insulin resistance, which limits the utilization of glucose and favors the increased utilization of free fatty acids for ketogenesis. These changes lead to a reduction in the production of high-energy phosphates and therefore to a metabolically inefficient heart.

Therefore, the metabolic changes in HF may favor for the progression and reducing functional capacity. Nevertheless, our findings suggest that the supply of fatty acids seems to initially exert a cardioprotective effect against the aggressions imposed by the obesity condition and pressure surges in the hyperlipidic models, since the HF diet did not probably promote systemic and myocardial insulin resistance (data not evaluated). In addition, it should be noted that the occurrence of heart failure was not observed in the evaluated experimental models; literature highlights the reduced use of fatty acids in heart failure [37], maintaining normal functioning of the heart.

On the other hand, the hyperglycemic model showed a cardiomyocyte contractile dysfunction. This finding corroborates Vasanji et al. [38], who demonstrated a reduction in the fractional shortening of animals fed a sucrose-rich diet, indicating cardiac contractility damage. Balderas-Villalobos et al. [39] reported that the consumption of sucrose in *Wistar* rats promoted a decreased $Ca^{2+}$ transient amplitude, although without modifications in the amount of $Ca^{2+}$ of sarcoplasmic reticulum (SR). These researchers suggested that alterations in the activity of L-type $Ca^{2+}$ channels or $RyR_2$ receptors may contribute to reduce $Ca^{2+}$ release and reuptake in SR via SERCA2a. In addition, Abel et al. [1] suggest that altered use of the myocardial substrate may be a mediator of subsequent contractile dysfunction. Within this context, when fatty acid levels are low and glucose concentrations are high, myocardial metabolism adapts to the use of glucose as an energetic substrate. This adaptive response is initially beneficial because it maintains adenosine triphosphate (ATP) levels in the face of decreased mitochondrial oxidative phosphorylation from fatty acids; however, this change in energy metabolism is not just a primary effect of cardiac remodeling, but it may, in fact, be a predictor of cardiac dysfunction [40]. Thus, the damage observed in the hyperglycemia model might be, among other factors, due to the higher glucose intake, which seems to cause some imbalance in intracellular homeostasis and, consequently, alter the mechanism of cardiac contraction and relaxation, as well as $Ca^{2+}$ handling.

Heart failure (HF) is the common endpoint of most heart disease, being one of the most important clinical challenges in the health field [41]. Obesity is an independent risk factor for the development of HF and this condition causes a series of hemodynamic changes, which aim to maintain the body's homeostasis, including increased cardiac output and decreased peripheral resistance, changing the cardiac morphology that predispose to left and right ventricular dysfunction. These results suggest that the damage in the cardiac morphology and function may lead to heart failure, even in the absence of other cardiac diseases [2, 6]. Our results indicate that no differences were found between the experimental groups, demonstrating the absence of HF. The absence of HF in the experimental models used in this study might be explained by the failure in the characterization of tissue congestion, RV hypertrophy, as well as systolic and diastolic dysfunction. In this sense, there were no morphological and functional adaptations, originating from the condition of obesity or excess sucrose, capable of predisposing to HF.

## Conclusion

The experimental diets based on high amounts of sugar, lard or a combination of both did not promote cardiac remodeling with predisposition to heart failure under conditions of obesity or excess sucrose. Nevertheless, excess sucrose causes cardiomyocyte contractility dysfunction associated with alterations in the myocyte sensitivity to intracellular $Ca^{2+}$.

## Acknowledgments

We are grateful to Priscilla Spadeto Altoé for their assistance.

## Author Contributions

**Conceptualization:** Amanda Martins Matias, Ana Paula Lima-Leopoldo, André Soares Leopoldo.

**Data curation:** Amanda Martins Matias, Vinícius Bermond Marques, Leonardo dos Santos, Aricia Leone Evangelista Monteiro de Assis, Breno Valentim Nogueira, Ana Paula Lima-Leopoldo.

**Formal analysis:** Amanda Martins Matias, Priscila Murucci Coelho, Vinícius Bermond Marques, Leonardo dos Santos, Aricia Leone Evangelista Monteiro de Assis, Ana Paula Lima-Leopoldo, André Soares Leopoldo.

**Investigation:** Amanda Martins Matias, Priscila Murucci Coelho, Vinícius Bermond Marques, Leonardo dos Santos, Aricia Leone Evangelista Monteiro de Assis, Breno Valentim Nogueira, Ana Paula Lima-Leopoldo, André Soares Leopoldo.

**Methodology:** Priscila Murucci Coelho, Leonardo dos Santos, Aricia Leone Evangelista Monteiro de Assis, Breno Valentim Nogueira.

**Project administration:** Amanda Martins Matias, André Soares Leopoldo.

**Resources:** Breno Valentim Nogueira, Ana Paula Lima-Leopoldo, André Soares Leopoldo.

**Software:** Vinícius Bermond Marques.

**Supervision:** Leonardo dos Santos, Breno Valentim Nogueira, Ana Paula Lima-Leopoldo, André Soares Leopoldo.

**Visualization:** Priscila Murucci Coelho, Vinícius Bermond Marques, Leonardo dos Santos, Aricia Leone Evangelista Monteiro de Assis, Ana Paula Lima-Leopoldo, André Soares Leopoldo.

**Writing – original draft:** Amanda Martins Matias, Priscila Murucci Coelho, Vinícius Bermond Marques, Leonardo dos Santos, Aricia Leone Evangelista Monteiro de Assis, Breno Valentim Nogueira, Ana Paula Lima-Leopoldo, André Soares Leopoldo.

**Writing – review & editing:** Amanda Martins Matias, Priscila Murucci Coelho, Vinícius Bermond Marques, Leonardo dos Santos, Aricia Leone Evangelista Monteiro de Assis, Breno Valentim Nogueira, Ana Paula Lima-Leopoldo, André Soares Leopoldo.

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
