## [Decision Letter · Decision Letter 0]

6 Nov 2019

PONE-D-19-28274

Hypercaloric diet models do not develop heart failure, but the excess sucrose promotes contractility dysfunction

PLOS ONE

Dear Mr Leopoldo,

Thank you for submitting your manuscript to PLOS ONE. After careful consideration, we feel that it has merit but does not fully meet PLOS ONE’s publication criteria as it currently stands. Therefore, we invite you to submit a revised version of the manuscript that addresses the points raised during the review process.

The authors need to address point-by-point responses to the reviewers' comments below before the manuscript is reconsidered for publication. To enhance the reproducibility of your results, we recommend that if applicable you deposit your laboratory protocols in protocols.io, where a protocol can be assigned its own identifier (DOI) such that it can be cited independently in the future. For instructions see: http://journals.plos.org/plosone/s/submission-guidelines#loc-laboratory-protocols

We look forward to receiving your revised manuscript.

Kind regards,

Tohru Minamino, M.D., Ph.D.

Academic Editor

PLOS ONE

Journal Requirements:

Reviewers' comments:

Reviewer's Responses to Questions

**Comments to the Author**

1. Is the manuscript technically sound, and do the data support the conclusions?

Reviewer #1: No

Reviewer #2: No

Reviewer #3: Partly

2. Has the statistical analysis been performed appropriately and rigorously? 

Reviewer #1: I Don't Know

Reviewer #2: Yes

Reviewer #3: Yes

3. Have the authors made all data underlying the findings in their manuscript fully available?

Reviewer #1: Yes

Reviewer #2: No

Reviewer #3: Yes

4. Is the manuscript presented in an intelligible fashion and written in standard English?

Reviewer #1: No

Reviewer #2: Yes

Reviewer #3: Yes

5. Review Comments to the Author

Reviewer #1: In this study the authors investigate the effect of HF and or HS sucrose diet on cardiac function and remodeling.

Overall the manuscript is poorly written, several sentences in particular in the introduction and discussion are either hard to understand or make no sense.

In the abstract it says 60 animals in total, 15 in each group. In the introduction it is down to 12/14/13/13 in each group, in the result part it is down to 5-8 animals. In particular when claiming absence of effect it is very important to have sufficient high n's. The entire study seems to be underpowered.

Overall, the manuscript contains very little data, maybe the data should have been combined with a article from 2018 from the same authors which seems to originate from the same study since the number of animals in each group stated in the introduction match the numbers in the current manuscript.

Reviewer #2: The manuscript by Matias et al. aims to develop a hypercaloric animal model of cardiac remodeling and predisposition to heart failure. The authors studied the effect of 3 different hypercaloric diets, one composed by high fat, other by high sucrose and the last one by the combination of high fat with high sucrose on heart and cardiomyocyte function. They found that none of the models studied promoted cardiac remodeling and predisposition to heart failure, however the sucrose diet causes cardiomyocyte contractility dysfunction. The manuscript is well written and the experiments were well design, although I believe it lacks mechanistic insights on why diets do not promote cardiac remodeling but high sucrose produces cardiomyocytes dysfunction.

1- Please describe the composition of the diets. This is an important factor as different diet composition might produce different disease phenotypes: heart failure, hypertension, among others. Note that several authors in the past has described the development of heart failure and hypertension with hypercaloric diet consumption (see e.g. Panchal and brown, 2011, Journal of Biomedicine and Biotechnology), and the composition of the diet might be of the factor that can contribute to the different results observed in the present manuscript. Also, diet composition, e.g. unsaturated fatty acids (UFA) can also contribute to the different results observed, see e.g. Carbone et al. 2017. JACC: Basic to Translational Science. Please discuss this.

2- Can the authors calculate the caloric intake of the animals? Was the caloric intake similar between groups?

3- How was the HFHsu administered? Was solid food? Or was sucrose administered in drinking water? Please describe the procedures, as this is quite important to interpret the results of the present manuscript. Usually, HFHSu models consume more sugar than fat and resemble more an HSu model than HF model.

4- Advanced glycation end-products (AGEs) can be endogenously formed as a consequence of a high dietary sugar intake and several lines of evidence suggest that AGEs are related to the development and progression of heart failure. I believe that the manuscript will truly benefit from the measurement of AGEs particularly because this might explain the differences between high fat and high sucrose diets.

5- Note, that the title and the conclusion of the manuscript depend totally on diet composition evaluated in the present study, please be more cautious on these conclusions.

Reviewer #3: This manuscript reports that the experimental hypercaloric models did not promote cardiac remodeling and predisposition to heart failure under conditions of obesity or excess sucrose. Nevertheless, excess sucrose causes cardiomyocyte contractility dysfunction associated with alterations in intracellular Ca2+. This work may add knowledge in the literature. The work in general is interesting, but there are important points and suggestion that should to be answered.

1 – The authors needs to organize the reference in the Introduction section (line 70 and 73).

2 - The authors describe in the Introduction Section that "….study aimed to investigate the cardiac remodeling process in an experimental model induced by different types of hypercaloric diets (high-fat (lard), high-sugar and the combination of both) and their effects on the development and progression of heart failure." I suggest changing the term ‘heart failure” for “...and their effects in the cardiac function”, because the experimental model do not is validated and do not develop the pathophysiology described;

3 – In the Results section, the diets association (HFHS) did induced any changes (figure 1), by the way, the HFD induces significantly changes in the cardiac parameters, So I would like to understand if the HS can interfere in the effect of HFD? This effect should be described in the Discussion section.

4 – The authors describe in the Discussion section that HS did cause morphological cardiac remodeling, but it is know the cardiotoxic effect of the high sugar. So I would like to understanding, did this diet works? Why we don not have the data of the metabolic profile? Moreover, I would like to suggest that the authors explain better, this absence of cardiotoxic effect in the HS.

5 – The author describe in the Discussion section a possible effect cardioprotetor in the initial stage of obesity, so I would like to suggest that this point be further explored and add in this section. Moreover, I suggest describing further details of the role of fatty acid reduction in heart failure.

6 – In the Discussion section, the authors explore the myocardial metabolic adapt to the use of glucose as an energetic substrate, however did not measure any parameter for this suggestion. So it is necessary describe further this point.

In addition, I would like to understand the effect of leptin in this HS experimental model due the intimal association of calcium concentration.

6. PLOS authors have the option to publish the peer review history of their article (what does this mean?). If published, this will include your full peer review and any attached files.

Reviewer #1: No

Reviewer #2: No

Reviewer #3: No

---

## [Author Response · Author response to Decision Letter 0]

20 Dec 2019

Manuscript Number: PONE-D-19-28274

Title: Hypercaloric diet models do not develop heart failure, but the excess sucrose promotes contractility dysfunction

Review Comments to the Author

Reviewer #1: 

1. In this study the authors investigate the effect of HF and or HS sucrose diet on cardiac function and remodeling. Overall the manuscript is poorly written, several sentences in particular in the introduction and discussion are either hard to understand or make no sense. 

Dear Reviewer, we appreciate the commentary, but we do not specifically understand what we need to change. In addition, the manuscript was sent to the English edition (English Certificate) and corrected by a specialized company. What do we need to be clearer?

2. In the abstract it says 60 animals in total, 15 in each group. In the introduction it is down to 12/14/13/13 in each group, in the result part it is down to 5-8 animals. In particular when claiming absence of effect it is very important to have sufficient high n's. The entire study seems to be underpowered. 

Dear Reviewer, we appreciate the commentary and agree that this information was confuse. Therefore, we believe a misconception has occurred and we have not expressed well about the number of animals. This information was added to the text, making it easier for readers to understand the total number of animals per group and per technique, since several techniques were used to respond to the initial study objective. Thus, each technique was performed with an appropriate number of animals to avoid absence of effect. For example, to morphological and histological analysis we used 5-8 animals from each group, because we would need the total left ventricle without going through any solution used in the cardiomyocyte technique. In addition, for the cardiomyocyte technique (5-6 animals each group), the animal's heart is fully utilized for cell dissociation and isolation. Therefore, we had to distribute the animals in different techniques. To general characteristics and to analysis of heart failure, we used all animals. It is noteworthy that current literature studies use this design strategy without asking for the effect.

The alteration was performed in the abstract. The correct sentence is “Thirty- day-old male Wistar rats (n = 52) were randomized into four groups: control (C), high sucrose (HS), high-fat (HF) and high-fat and sucrose (HFHS) for 20 weeks.” (Abstract, line 6)

3. Overall, the manuscript contains very little data, maybe the data should have been combined with a article from 2018 from the same authors which seems to originate from the same study since the number of animals in each group stated in the introduction match the numbers in the current manuscript. 

Dear Reviewer, we appreciate the commentary and disagree the actual manuscript contains very little data because in the previous study the aim was to investigate the effects of different dietary interventions on nutritional, metabolic, biochemical, hormonal, and cardiovascular profiles, as well as to add to development and characterization of an experimental model of obesity [1]. In the current study, our purpose was the development and characterization of an obesity experimental model from hypercaloric diets, which resulted in cardiac remodeling and predisposition to heart failure. In this sense, our would like to study the effects of these diets on heart. Generally, we divide the projects and study into several subprojects with different purposes, mainly in order to optimize the time and design of the experimental protocols, as well as the resources due to them lack by research funding agencies in Brazil. We believe this to occur frequently in world science, as it ends up optimizing the publication and discussion of the findings.

1. Matias AM, Estevam WM, Coelho PM, Haese D, Kobi JBBS, Lima-Leopoldo AP, Leopoldo AS. Differential Effects of High Sugar, High Lard or a Combination of Both on Nutritional, Hormonal and Cardiovascular Metabolic Profiles of Rodents. Nutrients. 2018 Aug 11;10(8). pii: E1071. doi: 10.3390/nu10081071. 

Reviewer #2: 

The manuscript by Matias et al. aims to develop a hypercaloric animal model of cardiac remodeling and predisposition to heart failure. The authors studied the effect of 3 different hypercaloric diets, one composed by high fat, other by high sucrose and the last one by the combination of high fat with high sucrose on heart and cardiomyocyte function. They found that none of the models studied promoted cardiac remodeling and predisposition to heart failure, however the sucrose diet causes cardiomyocyte contractility dysfunction. The manuscript is well written and the experiments were well design, although I believe it lacks mechanistic insights on why diets do not promote cardiac remodeling but high sucrose produces cardiomyocytes dysfunction. 

1- Please describe the composition of the diets. This is an important factor as different diet composition might produce different disease phenotypes: heart failure, hypertension, among others. Note that several authors in the past has described the development of heart failure and hypertension with hypercaloric diet consumption (see e.g. Panchal and brown, 2011, Journal of Biomedicine and Biotechnology), and the composition of the diet might be of the factor that can contribute to the different results observed in the present manuscript. Also, diet composition, e.g. unsaturated fatty acids (UFA) can also contribute to the different results observed, see e.g. Carbone et al. 2017. JACC: Basic to Translational Science. Please discuss this. 

Dear Reviewer, we appreciate the comment and we agree the composition of the diet might be of the factor that can contribute to the different results observed in the present manuscript. Our group published a specific study about the composition and experimental obesity model [1], in which it is possible to identify in detail the ingredients used and the nutritional value of diets (Nutrients 2018, 10(8), 1071; https://doi.org/10.3390/nu10081071), as detailed below: 

The experimental diets provided sufficient amounts of protein, vitamins, and minerals according to the Nutrient Requirements of Laboratory Animals [2]. The diets used in the current were formulated by Nutriave Alimentos® (Vitória, Espírito Santo, Brazil). The feed ingredients were blended, homogenized and extruded (Extru-Tech Extruder, Model E-750, Sabetha, KS, USA) in the form of pellets. Then, pellet was dried on a horizontal conveyor dryer (20 minutes, temperature: ± 70°C). The composition (g/kg) and nutrients of each experimental diet (%) is described in Table 1. 

Table 1. Composition and nutritional values of diets

Components (g/kg) Diets

 C HS HF HFHS

Corn 200 200 180 80

Rice 200 200 200 200

Bone meal 120 120 120 120

Sugar - 100 - 100

Soy oil 75 75 - -

Lard - - 200 200

Gluten 200 200 200 200

Salt 3.5 3.5 3.5 3.5

Mineral Mix** 35 35 30 30

Vitamin Mix** 16.5 16.5 16.5 16.5

Inert Material*** 150 50 50 50

Total (g) 1000 1000 1000 1000

Nutrient Composition (%) 

Protein 24.8 21.8 17.8 19.2

Carbohydrate 49.6 52.3 44.6 43.4

Lipids 25.6 25.9 37.6 37.4

Energy Density (Kcal/g) 3.55 3.65 4.59 4.49

Diets. C: normal rodent chow; HS: High-sugar; HF: high-fat; HFHS: high fat and high sugar. *Rats received diet with simple carbohydrate and water supplemented with sugar (300 g/L) in alternate weeks. In order to calculate the caloric intake of HS, the caloric value of the sugar diet (3.65 kcal/g) plus the caloric value of water intake with sugar (1.2 kcal/ml).

Unfortunately, we have not the composition of unsaturated fatty acids because we did not have resources at the time for the realization of these dosages, as well as equipment for this analysis. This suggestion will be applied in future studies. 

Nevertheless, as requested by Reviewer, the information about the composition of the diets was included in the text. The added sentence is “The diets used in current study were previously described by Amanda et al. [1], which provided sufficient amounts of protein, vitamins, and minerals” (Material and methods, Animal Care and Experimental design, page 4, lines 97-99)

1. Matias AM, Estevam WM, Coelho PM, Haese D, Kobi JBBS, Lima-Leopoldo AP, Leopoldo AS. Differential Effects of High Sugar, High Lard or a Combination of Both on Nutritional, Hormonal and Cardiovascular Metabolic Profiles of Rodents. Nutrients. 2018 Aug 11;10(8). pii: E1071. doi: 10.3390/nu10081071. 

2. Council, N.R. Nutrient Requirements of Swine; National Academies Press: Washington, DC, USA, 1998.

2- Can the authors calculate the caloric intake of the animals? Was the caloric intake similar between groups?

As requested by Reviewer, in order to analyze if dietary-induced obesity was associated with alterations in the nutritional behavior, the food consumption (FC) was measured daily and the calorie intake (CI) was calculated weekly by the average weekly FC × dietary energetic density. 

This information was previously published by our group [1]. However, the results (data not shown in the current study) showed that rats from C group had an approximate15.2%, 22.8% and 27.3% more daily food consumption (g) than HF, HFHS and HS groups, respectively, but the daily caloric intake was higher in the HS in relation to C and HFHS groups (HS: 92 ±2.0 vs. C: 79.2± 2.6and HFHS: 77.3 ± 1.7 kcal/day, p<0.05). In addition, there was different in FC between rats fed the HF diet compared with the HS group, since rats fed the HS diet consumed significantly less food (HG: 16.2 ± 0.4 vs. HF: 18.9 ± 0.7; p<0.05). Furthermore, HF presented a 12.4% increase in caloric intake than HFHS (p <0.05). There was no difference in the caloric intake of C compared to HF and HFHS (p> 0.05). 

1. Matias AM, Estevam WM, Coelho PM, Haese D, Kobi JBBS, Lima-Leopoldo AP, Leopoldo AS. Differential Effects of High Sugar, High Lard or a Combination of Both on Nutritional, Hormonal and Cardiovascular Metabolic Profiles of Rodents. Nutrients. 2018 Aug 11;10(8). pii: E1071. doi: 10.3390/nu10081071. 

3- How was the HFHsu administered? Was solid food? Or was sucrose administered in drinking water? Please describe the procedures, as this is quite important to interpret the results of the present manuscript. Usually, HFHSu models consume more sugar than fat and resemble more an HSu model than HF model. 

Dear Reviewer, we appreciate the commentary and as requested, the information about how was the HFHsu administered was mentioned below. This information was also previously published by our group [1].

Rats received diet (HFHS) fed a saturated high-fat diet and added simple sugar in the form of pellets. In addition, these rats received water supplemented with sugar (300 g/L) in alternate weeks”. In order to calculate the caloric intake of HS, the caloric value of the sugar diet (3.65 kcal/g) plus the caloric value of water intake with sugar (1.2 kcal/ml).

1. Matias AM, Estevam WM, Coelho PM, Haese D, Kobi JBBS, Lima-Leopoldo AP, Leopoldo AS. Differential Effects of High Sugar, High Lard or a Combination of Both on Nutritional, Hormonal and Cardiovascular Metabolic Profiles of Rodents. Nutrients. 2018 Aug 11;10(8). pii: E1071. doi: 10.3390/nu10081071. 

4- Advanced glycation end-products (AGEs) can be endogenously formed as a consequence of a high dietary sugar intake and several lines of evidence suggest that AGEs are related to the development and progression of heart failure. I believe that the manuscript will truly benefit from the measurement of AGEs particularly because this might explain the differences between high fat and high sucrose diets. 

Dear Reviewer, we appreciate the commentary and agree that the measurement of AGEs could be very important and enriched the work, but we don’t have any sample for this analysis. Really, 

as literature, this analysis is an emerging topic in the pathogenesis of metabolic diseases. AGEs can be ingested with high temperature processed foods, but also endogenously formed as a consequence of a high dietary sugar intake. We really like the suggestion and the idea will definitely be applied. Nevertheless, when this work was done, we didn´t have the prospect of carrying out the advanced glycation end-products (AGEs) in these animals because the initial aim of the current study was to investigate this study aimed to investigate the cardiac remodeling process in an experimental model induced by different types of hypercaloric diets (high-fat (lard), high-sugar and the combination of both) and their effects on the development and progression of heart failure. This suggestion will be applied in future studies and the projects in progress in our laboratory. 

5- Note, that the title and the conclusion of the manuscript depend totally on diet composition evaluated in the present study, please be more cautious on these conclusions. Dear Reviewer, we appreciate the commentary and agree that the title and the conclusion of the manuscript depend totally on diet composition evaluated in the present study, but we have the measure of diet composition as demonstrated in the answer 1. However, as requested we improved the conclusion but we still find the appropriate title based on our findings.

Title: Hypercaloric diet models do not develop heart failure, but the excess sucrose promotes contractility dysfunction

Conclusion: In summary, the experimental diets based on high amounts of sugar, lard or a combination of both did not promote cardiac remodeling with predisposition to heart failure under conditions of obesity or excess sucrose. Nevertheless, excess sucrose causes cardiomyocyte contractility dysfunction associated with alterations in the myocyte sensitivity to intracellular Ca2+.

Reviewer #3: 

This manuscript reports that the experimental hypercaloric models did not promote cardiac remodeling and predisposition to heart failure under conditions of obesity or excess sucrose. Nevertheless, excess sucrose causes cardiomyocyte contractility dysfunction associated with alterations in intracellular Ca2+. This work may add knowledge in the literature. The work in general is interesting, but there are important points and suggestion that should to be answered.

1 – The authors needs to organize the reference in the Introduction section (line 70 and 73). 

Dear Reviewer, we appreciate the commentary and we agree that the reference in the introduction section (line 70 and 73) were disorganized. As requested, this section was reorganized, and the reformulated section is described below:

Evidence from experimental studies indicates that high-fat feeding promotes cardiac contractile function damage [12, 13]. In our laboratory, myocardial dysfunction was demonstrated in basal conditions and physiological cardiac remodeling in obese rats [14-16]. In contrast, results on the effects of diets with high simple carbohydrate content (sucrose and/or fructose) are inconsistent. Sharma et al. [9] observed that hypertensive animals fed by fructose diet for eight weeks resulted in an increase in LV wall thickness and mortality, while Salie et al. [17]. (Introduction, page 3, line 70 and 73)

2 - The authors describe in the Introduction Section that "….study aimed to investigate the cardiac remodeling process in an experimental model induced by different types of hypercaloric diets (high-fat (lard), high-sugar and the combination of both) and their effects on the development and progression of heart failure." I suggest changing the term ‘heart failure” for “...and their effects in the cardiac function”, because the experimental model do not is validated and do not develop the pathophysiology described; 

As suggested by Reviewer, we changed this part of the introduction ("….study aimed to investigate the cardiac remodeling process in an experimental model induced by different types of hypercaloric diets (high-fat (lard), high-sugar and the combination of both) and their effects on the development and progression of heart failure”) and the sentence was rewritten as described below: 

Thus, this study aimed to investigate the cardiac remodeling process in an experimental model induced by different types of hypercaloric diets (high-fat (lard), high-sugar and the combination of both) and their effects in the cardiac function. (Introduction, page 4, line 81-84)

3 – In the Results section, the diets association (HFHS) did induced any changes (figure 1), by the way, the HFD induces significantly changes in the cardiac parameters, So I would like to understand if the HS can interfere in the effect of HFD? This effect should be described in the Discussion section. 

Dear Reviewer, we appreciate the commentary and as requested, we added in the discussion section how the HS diet can interfere in the effect of HFD, since the association between HS and HF did not promote changes in the weight of the total heart and left ventricle, as well as in the tibia ratios. Recalling that HF induced only significant changes in cardiac parameters in relation to HS. The discussion about this information is described below and it was added in the text:

Specifically, the HS did not promote cardiac remodeling, however, it was able to prevent the elevation on heart and LV weights in the HFHS, interfering in the effect of HF diet. According to literature, high sucrose intake [68% total energy sucrose diet, 69% total mass sucrose diet, or 20% sucrose solution] induced hyperinsulinemia without alterations in plasma glucose level in rats and mice [1-3]. Interestingly, our HS rats have not demonstrated alterations in glucose and insulin levels (data not shown). Several mechanisms have been postulated, including the prohypertrophic effects of insulin, insulin growth factor-1, and insulin resistance [4]. Initially the circulating insulin levels are increased, directly stimulating cardiomyocyte growth [5] and indirectly via binding to the insulin growth factor-1 receptor [6]. Chess et al. [7], analyzing the effects of dietary extremes (high carbohydrate and fat intake) on the remodeling process and heart failure, reported that high sugar intake causes hyperglycemia in the bloodstream and cardiotoxic effects. Increased glycemia leads to elevated serum insulin levels, which in cardiac tissue, induces increased protein synthesis of cardiomyocytes and, consequently, left ventricular hypertrophy. Sharma et al. [8] propose that carbohydrate intake, in particular sugar, associated or not with pressure overload may cause LV hypertrophy, via insulin receptor stimulation and activation of Akt/mTOR, proteins involved in protein signaling pathways. Thus, our results from HS diet suggest that the sugar intake was not able to affect the process of cardiac remodeling, probably due to the absence of hyperinsulinemia and elevation of glucose levels, as well as it was not promote cardiotoxic effect, indicating a cardioprotective effect alone or when associated with the HF diet. (Discussion, pages 11 and 12) 

The new references were added in the text.

1. Davidoff AJ, Mason MM, Davidson MB, Carmody MW, Hintz KK, Wold LE, Podolin DA, Ren J. Sucrose-induced cardiomyocyte dysfunction is both preventable and reversible with clinically relevant treatments. Am J Physiol Endocrinol Metab. 2004 May;286(5):E718-24.

2. Gonsolin D, Couturier K, Garait B, Rondel S, Novel-Chaté V, Peltier S, Faure P, Gachon P, Boirie Y, Keriel C, Favier R, Pepe S, Demaison L, Leverve X.High dietary sucrose triggers hyperinsulinemia, increases myocardial beta-oxidation, reduces glycolytic flux and delays post-ischemic contractile recovery. Mol Cell Biochem. 2007 Jan;295(1-2):217-28. Epub 2006 Aug 31.

3. Pang X, Zhao J, Zhang W, Zhuang X, Wang J, Xu R, Xu Z, Qu W. Antihypertensive effect of total flavones extracted from seed residues of Hippophae rhamnoides L. in sucrose-fed rats. J Ethnopharmacol. 2008 May 8;117(2):325-31. 

4. Phillips RA, Krakoff LR, Dunaif A, Finegood DT, Gorlin R, Shimabukuro S. Relation among left ventricular mass, insulin resistance, and blood pressure in nonobese subjects. J Clin Endocrinol Metab. 1998;83:4284–4288.

5. Hill DJ, Milner RD. Insulin as a growth factor. Pediatr Res. 1985;19:879–886.

6. Straus DS. Growth-stimulatory actions of insulin in vitro and in vivo. Endocr Rev. 1984;5:356–369. 

7. Chess DJ, Stanley WC. Role of diet and fuel overabundance in the development and progression of heart failure. Cardiovasc Res. 2008;79: 269-278. 

8. Sharma N, Okere IC, Duda MK, Chess DJ, O'Shea KM, Stanley WC. Potential impact of carbohydrate and fat intake on pathological left ventricular hypertrophy. Cardiovasc Res. 2007;73: 257-268.

4 – The authors describe in the Discussion section that HS did cause morphological cardiac remodeling, but it is know the cardiotoxic effect of the high sugar. So I would like to understanding, did this diet works? Why we don not have the data of the metabolic profile? Moreover, I would like to suggest that the authors explain better, this absence of cardiotoxic effect in the HS. 

Dear Reviewer, we appreciate the commentary and as requested and explained above, we added in the discussion section a possible explanation why the HS diet did not promote cardiotoxic effect. In addition, the metabolic profile (glucose and area under the curve for glucose; elevated in HF and HHFS groups) has been shown in our previous study [1], but the insulin levels (data not shown) were similar among the groups (C: 1.86±0.13; HS: 1.77±0.15, HF: 2.19±0.16 and HFHS: 2.39±0.23; p = 0.052). 

1. Matias AM, Estevam WM, Coelho PM, Haese D, Kobi JBBS, Lima-Leopoldo AP, Leopoldo AS. Differential Effects of High Sugar, High Lard or a Combination of Both on Nutritional, Hormonal and Cardiovascular Metabolic Profiles of Rodents. Nutrients. 2018 Aug 11;10(8). pii: E1071. doi: 10.3390/nu10081071. 

The discussion about this information is described below and it was added in the text:

Specifically, the HS did not promote cardiac remodeling, however, it was able to prevent the elevation on heart and LV weights in the HFHS, interfering in the effect of HF diet. According to literature, high sucrose intake [68% total energy sucrose diet, 69% total mass sucrose diet, or 20% sucrose solution] induced hyperinsulinemia without alterations in plasma glucose level in rats and mice [2-4]. Interestingly, our HS rats have not demonstrated alterations in glucose and insulin levels (data not shown). Several mechanisms have been postulated, including the prohypertrophic effects of insulin, insulin growth factor-1, and insulin resistance [5]. Initially the circulating insulin levels are increased, directly stimulating cardiomyocyte growth [6] and indirectly via binding to the insulin growth factor-1 receptor [7]. Chess et al. [8], analyzing the effects of dietary extremes (high carbohydrate and fat intake) on the remodeling process and heart failure, reported that high sugar intake causes hyperglycemia in the bloodstream and cardiotoxic effects. Increased glycemia leads to elevated serum insulin levels, which in cardiac tissue, induces increased protein synthesis of cardiomyocytes and, consequently, left ventricular hypertrophy. Sharma et al. [9] propose that carbohydrate intake, in particular sugar, associated or not with pressure overload may cause LV hypertrophy, via insulin receptor stimulation and activation of Akt/mTOR, proteins involved in protein signaling pathways. Thus, our results from HS diet suggest that the sugar intake was not able to affect the process of cardiac remodeling, probably due to the absence of hyperinsulinemia and elevation of glucose levels, as well as it was not promote cardiotoxic effect, indicating a cardioprotective effect alone or when associated with the HF diet. (Discussion, pages 11 and 12) 

The new references were added in the text.

.

2. Davidoff AJ, Mason MM, Davidson MB, Carmody MW, Hintz KK, Wold LE, Podolin DA, Ren J. Sucrose-induced cardiomyocyte dysfunction is both preventable and reversible with clinically relevant treatments. Am J Physiol Endocrinol Metab. 2004 May;286(5):E718-24.

3. Gonsolin D, Couturier K, Garait B, Rondel S, Novel-Chaté V, Peltier S, Faure P, Gachon P, Boirie Y, Keriel C, Favier R, Pepe S, Demaison L, Leverve X.High dietary sucrose triggers hyperinsulinemia, increases myocardial beta-oxidation, reduces glycolytic flux and delays post-ischemic contractile recovery. Mol Cell Biochem. 2007 Jan;295(1-2):217-28. Epub 2006 Aug 31.

4. Pang X, Zhao J, Zhang W, Zhuang X, Wang J, Xu R, Xu Z, Qu W. Antihypertensive effect of total flavones extracted from seed residues of Hippophae rhamnoides L. in sucrose-fed rats. J Ethnopharmacol. 2008 May 8;117(2):325-31. 

5. Phillips RA, Krakoff LR, Dunaif A, Finegood DT, Gorlin R, Shimabukuro S. Relation among left ventricular mass, insulin resistance, and blood pressure in nonobese subjects. J Clin Endocrinol Metab. 1998;83:4284–4288.

6. Hill DJ, Milner RD. Insulin as a growth factor. Pediatr Res. 1985;19:879–886.

7. Straus DS. Growth-stimulatory actions of insulin in vitro and in vivo. Endocr Rev. 1984;5:356–369. 

8. Chess DJ, Stanley WC. Role of diet and fuel overabundance in the development and progression of heart failure. Cardiovasc Res. 2008;79: 269-278. 

9. Sharma N, Okere IC, Duda MK, Chess DJ, O'Shea KM, Stanley WC. Potential impact of carbohydrate and fat intake on pathological left ventricular hypertrophy. Cardiovasc Res. 2007;73: 257-268.

5 – The author describe in the Discussion section a possible effect cardioprotetor in the initial stage of obesity, so I would like to suggest that this point be further explored and add in this section. Moreover, I suggest describing further details of the role of fatty acid reduction in heart failure. 

Dear Reviewer, we appreciate the commentary and as requested, we added in the discussion section a possible effect cardioprotective by reduction of utilization of fatty acids in myocardium in the initial stage of obesity, as well as we described with further details the role of fatty acid reduction in heart failure. The discussion about this information is described below and it was added in the text:

Free fatty acid oxidation is the major source of energy for the myocardium and up to 80 % of high-energy phosphates are produced, while the glucose metabolism provides the remaining quantity of energy. Glucose as glycogen are stored to be used during increased metabolic demands such as obesity and diabetes, since the glucose utilization is 20–30 % more metabolic efficient than free fatty acid oxidation in producing high-energy phosphates. In this sense, an energetic dysregulation play an important role in the pathophysiology of the failing heart. According Doesn’t et al. [1] a possible cause for these metabolic derangements in HF could be related to myocardial insulin resistance, which limits the utilization of glucose and favors the increased utilization of free fatty acids for ketogenesis. These changes lead to a reduction in the production of high-energy phosphates and therefore to a metabolically inefficient heart. Therefore, the metabolic changes in HF may favor for the progression and reducing functional capacity. Nevertheless, our findings suggest that this supply of fatty acids seems to initially exert a cardioprotective effect against the aggressions imposed by the obesity condition and pressure surges in the hyperlipidic models, since the HF diet did not probably promote systemic and myocardial insulin resistance (data not evaluated). In addition, it should be noted that the occurrence of heart failure was not observed in the evaluated experimental models; literature highlights the reduced use of fatty acids in heart failure [1], maintaining normal functioning of the heart. (Discussion, page 13, lines 8-26)

1. Doenst T, Nguyen TD, Abel ED. Cardiac metabolism in heart failure. Circ Res. 2013;113: 709-724.

6 – In the Discussion section, the authors explore the myocardial metabolic adapt to the use of glucose as an energetic substrate, however did not measure any parameter for this suggestion. So it is necessary describe further this point.

In addition, I would like to understand the effect of leptin in this HS experimental model due the intimal association of calcium concentration. 

Dear Reviewer, we appreciate the commentary and, really, we did not measure any parameter for evaluation of myocardial substrate as a mediator of subsequent contractile dysfunction in HS rats. As requested, the discussion about this was reformulated and it was added in the text:

In addition, Abel et al. [1] suggest that altered use of the myocardial substrate may be a mediator of subsequent contractile dysfunction. Within this context, when fatty acid levels are low and glucose concentrations are high, myocardial metabolism adapts to the use of glucose as an energetic substrate. This adaptive response is initially beneficial because it maintains adenosine triphosphate (ATP) levels in the face of decreased mitochondrial oxidative phosphorylation from fatty acids; however, this change in energy metabolism is not just a primary effect of cardiac remodeling, but it may, in fact, be a predictor of cardiac dysfunction [2]. (Discussion, page 14, lines 8-15)

1. Abel ED, Litwin SE, Sweeney G. Cardiac remodeling in obesity. Physiol Rev. 2008;88: 389-419.

2. Alfarano C, Foussal C, Lairez O, Calise D, Attané C, Anesia R, Daviaud D, Wanecq E, Parini A, Valet P, Kunduzova O. Transition from metabolic adaptation to maladaptation of the heart in obesity: role of apelin. Int J Obes (Lond). 2015 Feb;39(2):312-20. 

Leptin has been considered a key factor in the regulationof cardiovascular function, since since endogenous leptin production acts as aphysiological regulator of cardiovascular function, whereas hyperleptinemia or leptin resistance serves as pathophysiological indicator of cardiovascular diseases [3]. Leptin may reduce insulin release and enhance insulin sensitivity, consistent with elevated insulin. However, long-term hyperleptinemia commonly seen in human obesity may trigger leptin resistance, suppress insulin sensitivity and ultimately induce insulin resistance [4]. Nevertheless, our HS rats did not present hyperleptinemia that may be explained by the similar levels of adipose tissue compared to C (previously published data - 5). These results suggest that leptin did not impair the insulin release and, consequently, the myocardial glucose utilization.

3. Hintz KK, Aberle NS, Ren J. Insulin resistance induces hyperleptinemia, cardiac contractile dysfunction but not cardiac leptin resistance in ventricular myocytes. Int J Obes Relat Metab Disord. 2003 Oct;27(10):1196-203.

4. Ren J. Leptin and hyperleptinemia - from friend to foe for cardiovascular function. J Endocrinol. 2004 Apr;181(1):1-10.

5. Matias AM, Estevam WM, Coelho PM, Haese D, Kobi JBBS, Lima-Leopoldo AP, Leopoldo AS. Differential Effects of High Sugar, High Lard or a Combination of Both on Nutritional, Hormonal and Cardiovascular Metabolic Profiles of Rodents. Nutrients. 2018 Aug 11;10(8). pii: E1071. doi: 10.3390/nu10081071.

---

## [Decision Letter · Decision Letter 1]

2 Jan 2020

PONE-D-19-28274R1

Hypercaloric diet models do not develop heart failure, but the excess sucrose promotes contractility dysfunction

PLOS ONE

Dear Mr Leopoldo,

Thank you for submitting your manuscript to PLOS ONE. After careful consideration, we feel that it has merit but does not fully meet PLOS ONE’s publication criteria as it currently stands. Therefore, we invite you to submit a revised version of the manuscript that addresses the points raised during the review process.

Although the authors have answered some of the questions raised by the reviewers,  one reviewer still has some concerns that should be addressed. The authors need to give point-by-point responses to the reviewer's comments thoroughly before the manuscript is considered for publication.

To enhance the reproducibility of your results, we recommend that if applicable you deposit your laboratory protocols in protocols.io, where a protocol can be assigned its own identifier (DOI) such that it can be cited independently in the future. For instructions see: http://journals.plos.org/plosone/s/submission-guidelines#loc-laboratory-protocols

We look forward to receiving your revised manuscript.

Kind regards,

Tohru Minamino, M.D., Ph.D.

Academic Editor

PLOS ONE

Reviewers' comments:

Reviewer's Responses to Questions

**Comments to the Author**

1. If the authors have adequately addressed your comments raised in a previous round of review and you feel that this manuscript is now acceptable for publication, you may indicate that here to bypass the “Comments to the Author” section, enter your conflict of interest statement in the “Confidential to Editor” section, and submit your "Accept" recommendation.

Reviewer #2: (No Response)

2. Is the manuscript technically sound, and do the data support the conclusions?

Reviewer #2: Partly

3. Has the statistical analysis been performed appropriately and rigorously? 

Reviewer #2: Yes

4. Have the authors made all data underlying the findings in their manuscript fully available?

Reviewer #2: Yes

5. Is the manuscript presented in an intelligible fashion and written in standard English?

Reviewer #2: Yes

6. Review Comments to the Author

Reviewer #2: The authors have reviewed the manuscript entitled “Hypercaloric diet models do not develop heart failure, but the excess sucrose promotes contractility dysfunction” that aimed to develop a hypercaloric animal model of cardiac remodeling and predisposition to heart failure. The authors answered to some of the questions raised, however they failed to include the requested data into the manuscript and to add results that will give information on the mechanisms that may explain the differences between high fat and high sucrose diets.

Please see below my comments:

- Although some of the information requested by this reviewer was already included in a previous publication as stated by the authors “Matias AM, Estevam WM, Coelho PM, Haese D, Kobi JBBS, Lima-Leopoldo AP, Leopoldo AS. Differential Effects of High Sugar, High Lard or a Combination of Both on Nutritional, Hormonal and Cardiovascular Metabolic Profiles of Rodents. Nutrients. 2018 Aug 11;10(8). pii: E1071. doi: 10.3390/nu10081071.” e.g. composition of the diets, caloric intake, diet administration, this must be included in the present manuscript to allow the readers a full understand of the manuscript . This information is crucial to the manuscript as it main focus is to evaluate the effect of different diets on heart failure and cardiac contractility.

- Please include data of metabolic profile of the different diets, mainly insulin and glycemia values, specially in the context of the discussion of the mechanisms promoting cardiac remodeling (lines 263- 282 page 12)

- Please include descriptions for figure legends. e.g – Figure 1 – effect of different diet composition on…

7. PLOS authors have the option to publish the peer review history of their article (what does this mean?). If published, this will include your full peer review and any attached files.

Reviewer #2: No

---

## [Author Response · Author response to Decision Letter 1]

21 Jan 2020

The response to Reviewers was attached in the attach files and described below.

Manuscript Number: PONE-D-19-28274R1

Title: Hypercaloric diet models do not develop heart failure, but the excess sucrose promotes contractility dysfunction

Review Comments to the Author

Reviewer #2: 

The authors have reviewed the manuscript entitled “Hypercaloric diet models do not develop heart failure, but the excess sucrose promotes contractility dysfunction” that aimed to develop a hypercaloric animal model of cardiac remodeling and predisposition to heart failure. The authors answered to some of the questions raised, however they failed to include the requested data into the manuscript and to add results that will give information on the mechanisms that may explain the differences between high fat and high sucrose diets.

Please see below my comments:

1 - Although some of the information requested by this reviewer was already included in a previous publication as stated by the authors “Matias AM, Estevam WM, Coelho PM, Haese D, Kobi JBBS, Lima-Leopoldo AP, Leopoldo AS. Differential Effects of High Sugar, High Lard or a Combination of Both on Nutritional, Hormonal and Cardiovascular Metabolic Profiles of Rodents. Nutrients. 2018 Aug 11;10(8). pii: E1071. doi: 10.3390/nu10081071.” e.g. composition of the diets, caloric intake, diet administration, this must be included in the present manuscript to allow the readers a full understand of the manuscript. This information is crucial to the manuscript as it main focus is to evaluate the effect of different diets on heart failure and cardiac contractility.

Dear Reviewer, we appreciate the comment and as requested the composition of the diets, caloric intake and diet administration were added in the text. In addition, the Table 1 was added in the current manuscript. 

The added sentence about the composition of the diets was “The experimental diets provided sufficient amounts of protein, vitamins and minerals according to the Nutrient Requirements for Laboratory Animals. The diets used in the current study were formulated by Nutriave Alimentos® (Vitória, Espírito Santo, Brazil). The feed ingredients were blended, homogenized and extruded (Extru-Tech Extruder, Model E-750, Sabetha, KS, USA) in the form of pellets. Then, the pellets were dried on a horizontal conveyor dryer (20 minutes, temperature: ±70° C). The composition (g/kg) and nutrients for each experimental diet (%) are described in Table 1.” (Material and Methods, Animal care and Experimental design, page 5, line 3-9)

Table 1. Composition and nutritional values of diets

Components (g/kg) Diets

 C HS HF HFHS

Corn 200 200 180 80

Rice 200 200 200 200

Bone meal 120 120 120 120

Sugar - 100 - 100

Soy oil 75 75 - -

Lard - - 200 200

Gluten 200 200 200 200

Salt 3.5 3.5 3.5 3.5

Mineral Mix** 35 35 30 30

Vitamin Mix** 16.5 16.5 16.5 16.5

Inert Material*** 150 50 50 50

Total (g) 1000 1000 1000 1000

Nutrient Composition (%) 

Protein 24.8 21.8 17.8 19.2

Carbohydrate 49.6 52.3 44.6 43.4

Lipids 25.6 25.9 37.6 37.4

Energy Density (Kcal/g) 3.55 3.65 4.59 4.49

Diets. C: normal rodent chow; HS: High-sugar; HF: high-fat; HFHS: high fat and high sugar. *Rats received diet with simple carbohydrate and water supplemented with sugar (300 g/L) in alternate weeks. In order to calculate the caloric intake of HS, the caloric value of the sugar diet (3.65 kcal/g) plus the caloric value of water intake with sugar (1.2 kcal/ml).

As requested by Reviewer, in order to analyze if dietary-induced obesity was associated with alterations in the nutritional behavior, the food consumption (FC) was measured daily and the calorie intake (CI) was calculated weekly by the average weekly FC × dietary energetic density. These information and description of results were added in the Table 2 and results section. (Results, page 9, line 1-9)

The added sentence about the nutritional behavior was “The nutritional profile of rats is summarized in Table 2. The C rats had an approximately 17.9%, 29.6% and 37.6% greater daily food consumption (g) than the HF, HFHS and HS groups, respectively, but the daily caloric intake was higher in the HS group in relation to the C and HFHS groups (HS: 92.1 ± 2.1 vs. C: 79.2 ± 2.6 and HFHS: 77.3 ± 1.8 kcal/day, p < 0.05). In addition, there was a difference in FC between rats fed the HF diet compared with the HS group since rats fed the HS diet consumed significantly less food (HG: 16.2 ± 0.5 vs. HF:18.9 ± 0.7; p < 0.05). Furthermore, HF presented a 12.4% increase in caloric intake over HFHS (p < 0.05). There was no difference in the caloric intake of C compared to HF and HFHS (p > 0.05).”

Table 2. General Characteristics 

Variables Experimental Groups

 C HS HF HFHS

IBW (g) 107 ± 3 110 ± 3 111 ± 3 110 ± 4

FBW (g) 533 ± 17 538 ± 13# 649 ± 34* 616 ± 22

BW gain (g) 426 ± 17 428 ± 12# 538 ± 32* 506 ± 20

Epididymal fat pad (g) 11.2 ± 0.6 10.5 ± 0.7 13.1 ± 0.9 13.5 ± 1.0

Visceral fat pad (g) 11.4 ± 0.6 10.6 ± 0.6#& 18.5 ± 1.4* 15.9 ± 1.4*

Retroperitoneal fat pad (g) 21.9 ± 1.0 21.8 ± 2.0#& 40.4 ± 4.5* 34.4 ± 2.2*

Body fat (g) 44.5 ± 1.6 42.9 ± 0.6#& 72.0 ± 6.3* 63.8 ± 4.3*

Adiposity index (%) 8.3 ± 0.2 7.9 ± 0.3#& 10.9 ± 0.5* 10.2 ± 0.3*

Food consumption (g/day) 22.3 ± 0.7 16.2 ± 0.5 18.9 ± 0.7 17.2 ± 0.4

Caloric intake (kcal/day) 79.2 ± 2.6 92.1 ± 2.1 86.9 ± 3.5 77.3 ± 1.8

Glucose (mg/dL) 108 ± 2 112 ± 3 115 ± 4 115 ± 3

Insulin (ng/mL) 1.86 ± 0.13 1.77 ± 0.15 2.19 ± 0.16 2.39 ± 0.23

Data are presented as the mean ± SEM. Control diet - (C; n=12); high-sugar diet - (HS; n=14); high-fat diet - (HF; n=13), and high-fat and high-sugar diet (HFHS; n=13). IBW: initial body weight; FBW: final body weight; BW: body weight. One-way ANOVA for independent samples followed by Tukey post hoc test. p < 0.05 vs. * C; # HF vs. HS; & HFHS vs. HS.

Dear Reviewer, we appreciate the commentary and as requested, the information about how was the HFHsu administered was mentioned below and added in the text. 

All animals had free access to water and chow (40 g/day). To analyze whether dietary-induced obesity was associated with alterations in nutritional behavior, food consumption (FC) was measured daily. Calorie intake (CI) was calculated weekly by the average weekly FC × dietary energetic density. The HS group had water supplemented with sugar (300 g/l) in alternate weeks. For the calculation of the caloric intake of the HS group, the caloric energy from the water supplemented with sugar was also quantified (1.2 kcal/mL consumed). (Material and Methods, Animal care and Experimental design, page 4, line 2-7)

2 - Please include data of metabolic profile of the different diets, mainly insulin and glycemia values, specially in the context of the discussion of the mechanisms promoting cardiac remodeling (lines 263- 282 page 12)

Dear Reviewer, we appreciate the commentary and as requested, we included data of metabolic profile (insulin and glucose) of the different diets. This information was added in the Table 2, results and discussion sections. The sentence added was described below.

The metabolic profile parameters including glucose (p=0.26) and insulin (p=0.06) were similar among the groups (Table 2). (Results section, page 9, lines 9-10)

Table 2. General Characteristics 

Variables Experimental Groups

 C HS HF HFHS

IBW (g) 107 ± 3 110 ± 3 111 ± 3 110 ± 4

FBW (g) 533 ± 17 538 ± 13# 649 ± 34* 616 ± 22

BW gain (g) 426 ± 17 428 ± 12# 538 ± 32* 506 ± 20

Epididymal fat pad (g) 11.2 ± 0.6 10.5 ± 0.7 13.1 ± 0.9 13.5 ± 1.0

Visceral fat pad (g) 11.4 ± 0.6 10.6 ± 0.6#& 18.5 ± 1.4* 15.9 ± 1.4*

Retroperitoneal fat pad (g) 21.9 ± 1.0 21.8 ± 2.0#& 40.4 ± 4.5* 34.4 ± 2.2*

Body fat (g) 44.5 ± 1.6 42.9 ± 0.6#& 72.0 ± 6.3* 63.8 ± 4.3*

Adiposity index (%) 8.3 ± 0.2 7.9 ± 0.3#& 10.9 ± 0.5* 10.2 ± 0.3*

Food consumption (g/day) 22.3 ± 0.7 16.2 ± 0.5 18.9 ± 0.7 17.2 ± 0.4

Caloric intake (kcal/day) 79.2 ± 2.6 92.1 ± 2.1 86.9 ± 3.5 77.3 ± 1.8

Glucose (mg/dL) 108 ± 2 112 ± 3 115 ± 4 115 ± 3

Insulin (ng/mL) 1.86 ± 0.13 1.77 ± 0.15 2.19 ± 0.16 2.39 ± 0.23

Data are presented as the mean ± SEM. Control diet - (C; n=12); high-sugar diet - (HS; n=14); high-fat diet - (HF; n=13), and high-fat and high-sugar diet (HFHS; n=13). IBW: initial body weight; FBW: final body weight; BW: body weight. One-way ANOVA for independent samples followed by Tukey post hoc test. p < 0.05 vs. * C; # HF vs. HS; & HFHS vs. HS.

Specifically, the HS did not promote cardiac remodeling, however, it was able to prevent the elevation on heart and LV weights in the HFHS, interfering in the effect of HF diet. According to literature, high sucrose intake [68% total energy sucrose diet, 69% total mass sucrose diet, or 20% sucrose solution] induced hyperinsulinemia without alterations in plasma glucose level in rats and mice [28-30]. Interestingly, our HS rats have not demonstrated alterations in glucose and insulin levels (Table 2). Several mechanisms have been postulated, including the prohypertrophic effects of insulin, insulin growth factor-1, and insulin resistance [31]. Initially the circulating insulin levels are increased, directly stimulating cardiomyocyte growth [32] and indirectly via binding to the insulin growth factor-1 receptor [33]. Chess et al. [26], analyzing the effects of dietary extremes (high carbohydrate and fat intake) on the remodeling process and heart failure, reported that high sugar intake causes hyperglycemia in the bloodstream and cardiotoxic effects. Increased glycemia leads to elevated serum insulin levels, which in cardiac tissue, induces increased protein synthesis of cardiomyocytes and, consequently, left ventricular hypertrophy. Sharma et al. [9] propose that carbohydrate intake, in particular sugar, associated or not with pressure overload may cause LV hypertrophy, via insulin receptor stimulation and activation of Akt/mTOR, proteins involved in protein signaling pathways. Thus, our results from HS diet suggest that the sugar intake was not able to affect the process of cardiac remodeling, probably due to the absence of hyperinsulinemia and elevation of glucose levels, as well as it was not promote cardiotoxic effect, indicating a cardioprotective effect alone or when associated with the HF diet. (Discussion section, page 13, line 12)

3 - Please include descriptions for figure legends. e.g – Figure 1 – effect of different diet composition on…

Dear Reviewer, we appreciate the commentary and as requested, we included the descriptions for all figures (Figs. 1-4). 

The description of figures is described below, and it was added in the text:

Figure 1. Effect of different diet composition on cardiac remodeling. Data are shown as mean ± SEM. Control diet - (C; n=5); high-sugar diet - (HS; n=8); high-fat diet - (HF; n=5), and high-fat and high-sugar diet (HFHS; n=6). HW: heart weight; LW: left weight. p < 0.05 vs. # HF vs. HS. One-way analysis of variance (ANOVA) followed by the Tukey post hoc test.

Figure 2. Histological study in myocardium from control diet - (C; n=5); high-sugar diet - (HS; n=8); high-fat diet - (HF; n=5), and high-fat and high-sugar diet (HFHS; n=6) after 20 weeks. (A) cross sectional area (CSA). (B): interstitial collagen of myocardium; representative picrosirius red-stained left ventricle (LV) section. Arrows: interstitial collagen. Data are shown as mean ± SEM. One-way analysis of variance (ANOVA) followed by the Tukey post hoc test. p < 0.05 vs. * C; & HFHS vs. HS; §HFHS vs. HF. 

Figure 3. Effect of different diet composition on parameters of heart failure. Data are shown as mean ± SEM. Control diet - (C; n=12); high-sugar diet - (HS; n=14); high-fat diet - (HF; n=13), and high-fat and high-sugar diet (HFHS; n=13). RV: right ventricle. (A) Lung weight/body weight ratio. (B) Right lung weight/body weight ratio. (C) RV weight/body weight. (D) Lung water content (C = 5; HS = 8; HF = 5, and HFHS = 6). One-way analysis of variance (ANOVA) followed by the Tukey post hoc test.

Figure 4. Effect of different diet composition on contractile function and calcium transients of left ventricular cardiomyocytes from control diet (C; n= 5; cells= 71), high-sugar diet (HS; n=6, cells= 115), high-fat diet (HF; n= 6, cells= 106) and high-fat and high-sugar diet (HFHS; n=5, cells= 81). Data are shown as mean ± SEM. (A) Representative contraction traces obtained from the cardiomyocytes of rats: (B) Cell shortening expressed as % of resting cell length. (C) Time to 50% of contraction. (D) Time to 50% of relaxation. (E) Amplitude of transients. (F) Time to peak. (G) Time to from peak transient to half resting value p < 0.05 vs. * C; # HS vs. HF; & HFHS vs. HS. One-way analysis of variance (ANOVA) followed by the Tukey post hoc test.

---

## [Decision Letter · Decision Letter 2]

23 Jan 2020

PONE-D-19-28274R2

Hypercaloric diet models do not develop heart failure, but the excess sucrose promotes contractility dysfunction

PLOS ONE

Dear Mr Leopoldo,

Thank you for submitting your manuscript to PLOS ONE. After careful consideration, we feel that it has merit but does not fully meet PLOS ONE’s publication criteria as it currently stands. Therefore, we invite you to submit a revised version of the manuscript that addresses the points raised during the review process.

Although the authors have addressed some of the questions raised by the reviewer, the reviewer still has two additional comments below. The authors need to give point-by-point responses to the reviewer's comments before the manuscript is considered for publication. 

 To enhance the reproducibility of your results, we recommend that if applicable you deposit your laboratory protocols in protocols.io, where a protocol can be assigned its own identifier (DOI) such that it can be cited independently in the future. For instructions see: http://journals.plos.org/plosone/s/submission-guidelines#loc-laboratory-protocols

We look forward to receiving your revised manuscript.

Kind regards,

Tohru Minamino, M.D., Ph.D.

Academic Editor

PLOS ONE

Reviewers' comments:

Reviewer's Responses to Questions

**Comments to the Author**

1. If the authors have adequately addressed your comments raised in a previous round of review and you feel that this manuscript is now acceptable for publication, you may indicate that here to bypass the “Comments to the Author” section, enter your conflict of interest statement in the “Confidential to Editor” section, and submit your "Accept" recommendation.

Reviewer #2: (No Response)

2. Is the manuscript technically sound, and do the data support the conclusions?

Reviewer #2: Partly

3. Has the statistical analysis been performed appropriately and rigorously? 

Reviewer #2: Yes

4. Have the authors made all data underlying the findings in their manuscript fully available?

Reviewer #2: No

5. Is the manuscript presented in an intelligible fashion and written in standard English?

Reviewer #2: Yes

6. Review Comments to the Author

Reviewer #2: The authors have reviewed the manuscript entitled “Hypercaloric diet models do not develop heart failure, but the excess sucrose promotes contractility dysfunction” that aimed to develop a hypercaloric animal model of cardiac remodeling and predisposition to heart failure. The authors have answered adequately to some of the questions raised however I have 2 additional comments:

1- I really don’t understand how the animals HFHS that have a decreased caloric intake in comparison with the controls gain weight and have an increased body fat and adiposity index compared with controls. (table 2)

2- Please include in methods section, the methods to analyze insulin and glucose.

7. PLOS authors have the option to publish the peer review history of their article (what does this mean?). If published, this will include your full peer review and any attached files.

Reviewer #2: No

---

## [Author Response · Author response to Decision Letter 2]

24 Jan 2020

Manuscript Number: PONE-D-19-28274R2

Title: Hypercaloric diet models do not develop heart failure, but the excess sucrose promotes contractility dysfunction

Review Comments to the Author

Reviewer #2: The authors have reviewed the manuscript entitled “Hypercaloric diet models do not develop heart failure, but the excess sucrose promotes contractility dysfunction” that aimed to develop a hypercaloric animal model of cardiac remodeling and predisposition to heart failure. The authors have answered adequately to some of the questions raised however I have 2 additional comments:

1- I really don’t understand how the animals HFHS that have a decreased caloric intake in comparison with the controls gain weight and have an increased body fat and adiposity index compared with controls. (table 2)

Dear Reviewer, we appreciate the commentary and as requested, the explanation about how the animals HFHS have a decreased caloric intake, but higher body fat and adiposity index is related to feed efficiency, which it is the ability to transform consumed calories into body weight, was determined by following the formula: mean body weight gain (g)/total calorie intake (kcal). In this sense, the feed efficiency (%) was higher in the HF (14.3%) and HFHS groups (20.8%) than in C (data included in the Table 2) as demonstrated (HFHS: 4.64 ± 0.09 and HF: 4.39 ± 0.09 vs. C: 3.84 ± 0.08; p < 0.05). The elevation in body weight and adiposity in the hyperlipidic experimental models (HF) increased with sugar, refers to caloric intake and food efficiency which, generally, lead to a greater availability of calories and hyperphagia [1,2]. Even without an increase in caloric intake, it is possible to develop obesity, since the change in nutrient composition influences the efficiency of food utilization, thus leading to an increase in fat storage per calorie consumed. The elevation in body fat may be due to an increase in the caloric density of the diet, which may lead to a higher total caloric intake or an increase in the intake of a given macronutrient [3]. Although they had a similar total caloric intake, the HF and HFHS groups consumed 61% and 43% more calories from fat, respectively, than the C animals did. Therefore, the elevation of feed efficiency in HFHS could explain how the animals HFHS had an increased body fat and adiposity index compared with C rats, despite the absence of alterations in caloric intake among HFHS and HF in relation to C group (Table 2).

1. Dourmashkin J, Chang G, Gayles E, Hill J, Fried S, Julien C, Leibowitz S. Different forms of obesity as a function of diet composition. Int. J. Obes. 2005;29:1368–1378.

2. White S, Cercato LM, Araujo D, Souza LA, Soares AF, Barbosa APO, et al. Modelo de obesidade induzida por dieta hiperlipidica e associada à resistência à ação da insulina e intolerância a glicose. Arq Bras Endocrinol Metab 2013;57:339–345.

3. Pereira-Lancha LO, Campos-Ferraz PL, Lancha AH. Obesity: Considerations about etiology, metabolism, and the use of experimental models. Diabetes Metab Syndr Obes. 2012;5:75–87.

The added sentences and Table 2 were detailed below:

In order to analyze if dietary-induced obesity was associated with alterations in the nutritional behavior, the food consumption (FC) was measured daily and the calorie intake (CI) was calculated weekly by the average weekly FC × dietary energetic density, as well as the feed efficiency to show the higher body fat and adiposity index in HF and HFHS in comparison to C group (Table 2). These information and description of results were added in the Table 2 and results section. (Results, page 9, lines 1-11)

The added sentence about the nutritional behavior was “The nutritional profile of rats is summarized in Table 2. The C rats had an approximately 17.9%, 29.6% and 37.6% greater daily food consumption (g) than the HF, HFHS and HS groups, respectively, but the daily caloric intake was higher in the HS group in relation to the C and HFHS groups (HS: 92.1 ± 2.1 vs. C: 79.2 ± 2.6 and HFHS: 77.3 ± 1.8 kcal/day, p < 0.05). In addition, there was a difference in FC between rats fed the HF diet compared with the HS group since rats fed the HS diet consumed significantly less food (HG: 16.2 ± 0.5 vs. HF:18.9 ± 0.7; p < 0.05). Furthermore, HF presented a 12.4% increase in caloric intake over HFHS (p < 0.05). There was no difference in the caloric intake of C compared to HF and HFHS (p > 0.05). While the feed efficiency (%) was higher in the HF (14.3%) and HFHS groups (20.8%) than in C (Table 2), there was a lower feed efficiency in HS rats than C (HS: 3.32 ± 0.05 vs. C: 3.84 ± 0.08; p < 0.05).” (Results, pages 9 and 10, lines 1-11)

Table 2. General Characteristics 

Variables Experimental Groups

 C HS HF HFHS

IBW (g) 107 ± 3 110 ± 3 111 ± 3 110 ± 4

FBW (g) 533 ± 17 538 ± 13# 649 ± 34* 616 ± 22

BW gain (g) 426 ± 17 428 ± 12# 538 ± 32* 506 ± 20

Epididymal fat pad (g) 11.2 ± 0.6 10.5 ± 0.7 13.1 ± 0.9 13.5 ± 1.0

Visceral fat pad (g) 11.4 ± 0.6 10.6 ± 0.6#& 18.5 ± 1.4* 15.9 ± 1.4*

Retroperitoneal fat pad (g) 21.9 ± 1.0 21.8 ± 2.0#& 40.4 ± 4.5* 34.4 ± 2.2*

Body fat (g) 44.5 ± 1.6 42.9 ± 0.6#& 72.0 ± 6.3* 63.8 ± 4.3*

Adiposity index (%) 8.3 ± 0.2 7.9 ± 0.3#& 10.9 ± 0.5* 10.2 ± 0.3*

Food consumption (g/day) 22.3 ± 0.7 16.2 ± 0.5*# 18.9 ± 0.7* 17.2 ± 0.4*

Caloric intake (kcal/day) 79.2 ± 2.6 92.1 ± 2.1*& 86.9 ± 3.5 77.3 ± 1.8α

Feed efficiency (%) 3.84 ± 0.08 3.32 ± 0.05* 4.39 ± 0.09*# 4.64 ± 0.09*&

Glucose (mg/dL) 108 ± 2 112 ± 3 115 ± 4 115 ± 3

Insulin (ng/mL) 1.86 ± 0.13 1.77 ± 0.15 2.19 ± 0.16 2.39 ± 0.23

Data are presented as the mean ± SEM. Control diet - (C; n=12); high-sugar diet - (HS; n=14); high-fat diet - (HF; n=13), and high-fat and high-sugar diet (HFHS; n=13). IBW: initial body weight; FBW: final body weight; BW: body weight. One-way ANOVA for independent samples followed by Tukey post hoc test. p < 0.05 vs. * C; # HF vs. HS; & HFHS vs. HS, α HF vs. HFHS.

Dear Reviewer, the information about how was the analyze of feed efficiency is mentioned below and added in the text. 

All animals had free access to water and chow (40 g/day). To analyze whether dietary-induced obesity was associated with alterations in nutritional behavior, food consumption (FC) was measured daily. Calorie intake (CI) was calculated weekly by the average weekly FC × dietary energetic density. Feed efficiency (FE), the ability to transform consumed calories into body weight, was determined by following the formula: mean body weight gain (g)/total calorie intake (kcal). The HS group had water supplemented with sugar (300 g/l) in alternate weeks. For the calculation of the caloric intake of the HS group, the caloric energy from the water supplemented with sugar was also quantified (1.2 kcal/mL consumed). (Material and Methods, Animal care and Experimental design, page 4, line 2-10)

2- Please include in methods section, the methods to analyze insulin and glucose.

Dear Reviewer, we appreciate the comment and as requested the methods to analyze insulin and glucose was added in the methods section.

The added sentence is described below:

Metabolic and Hormonal Measurements

After 20 weeks, the animals were subjected to 12–15 h of fasting, and blood samples were collected in dry tubes. The serum was separated by centrifugation at 10,000 rpm for 10 min. (Heraeus Megafuge 16R Centrifuge, Thermo Scientific, Massachusetts, USA) and stored at −80 °C for subsequent analysis (Coldlab Ultra Freezer CL374-86V, Piracicaba, São Paulo, Brazil). Serum glucose concentration was measured using specific kit (Bioclin Bioquímica®, Belo Horizonte, Minas Gerais, Brazil and Synermed do Brasil Ltda., São Paulo, Brazil) and analyzed by automated biochemical equipment BS-200 (Mindray do Brasil-Comércio and Distribuição de Equipamentos Médicos Ltda., São Paulo, Brazil). Insulin was determined using an enzyme-linked immunosorbent assay (ELISA) using specific kit (Linco Research Inc., St. Louis, MO, USA). The reading was carried out using a microplate reader (Asys Expert Plus Microplate Reader, Cambourne, Cambridge, UK). (Material and Methods, Metabolic and Hormonal Measurements, page 6, lines 1-11)

---

## [Editor Report · Decision Letter 3]

27 Jan 2020

Hypercaloric diet models do not develop heart failure, but the excess sucrose promotes contractility dysfunction

PONE-D-19-28274R3

Dear Dr. Leopoldo,

We are pleased to inform you that your manuscript has been judged scientifically suitable for publication and will be formally accepted for publication once it complies with all outstanding technical requirements.

With kind regards,

Tohru Minamino, M.D., Ph.D.

Academic Editor

PLOS ONE

---

## [Editor Report · Acceptance letter]

31 Jan 2020

PONE-D-19-28274R3 

Hypercaloric diet models do not develop heart failure, but the excess sucrose promotes contractility dysfunction 

Dear Dr. Soares Leopoldo:

I am pleased to inform you that your manuscript has been deemed suitable for publication in PLOS ONE. Congratulations! Your manuscript is now with our production department. 

With kind regards,

on behalf of

Professor Tohru Minamino 

Academic Editor

PLOS ONE